# Integer Programming for Generalized Causal Bootstrap Designs

**Jennifer Brennan** [1]  **Sébastien Lahaie** [1]  **Adel Javanmard** [1,2]  **Nick Doudchenko** [3]  **Jean Pouget-Abadie** [1]

## Abstract

In experimental causal inference, we distinguish between two sources of uncertainty: design uncertainty, due to the treatment assignment mechanism, and sampling uncertainty, when the sample is drawn from a super-population. This distinction matters in settings with small fixed samples and heterogeneous treatment effects, as in geographical experiments. The standard bootstrap procedure most often used by practitioners primarily estimates sampling uncertainty, and the causal bootstrap procedure, which accounts for design uncertainty, was developed for the completely randomized design and the difference-in-means estimator, whereas non-standard designs and estimators are often used in these low-power regimes. We address this gap by proposing an integer program which computes numerically the worst-case copula used as an input to the causal bootstrap method, in a wide range of settings. Specifically, we prove the asymptotic validity of our approach for unconfounded, conditionally unconfounded, and individualistic with bounded confoundedness assignments, as well as generalizing to any linear-in-treatment and quadratic-in-treatment estimators. We demonstrate the refined confidence intervals achieved through simulations of small geographical experiments.

## 1. Introduction

Randomized controlled trials (RCTs) are used to estimate causal effects across the sciences, social sciences, and industry. An important complement to estimating causal effects is providing valid uncertainty quantification, which enables statistically sound decision-making between tested policies. We distinguish between two types of uncertainty: *sampling uncertainty* and *design uncertainty*. *Sampling uncertainty*

captures the uncertainty associated with inferring statistics of a population given a random sample of that population. *Design uncertainty*, by contrast, captures the uncertainty associated with the random assignment of units to treatment and control conditions. Design uncertainty is associated with the *finite population model* of causal inference, in which the universe consists of exactly $N$ units. Some units are observed under the treatment condition while others are observed under the control. The objective in the finite population model is to estimate the average difference between treated outcomes and control outcomes for these $N$ units, typically in the absence of distributional assumptions on these outcomes. Uncertainty arises from the fact that each unit is only observed under the treated *or* controlled condition, so that any statistic derived from RCT data will vary slightly from one randomization to another.

Sampling uncertainty has received extensive attention in the statistics literature. Well-known methods to estimate sampling uncertainty include the bootstrap (Efron, 1979), in which random draws from the sample mimic the population distribution, and Wald-type confidence intervals, which rely on asymptotic normality of the sample statistic. The methodological toolkit available to quantify design uncertainty is considerably more sparse (Imbens & Menzel, 2021; Aronow et al., 2014). For reasons explained in Appendix 2.1, methods to estimate sampling uncertainty are typically conservative for design uncertainty. The gap widens for experiments with heterogeneous effects, small sample sizes, and a large fraction of the population included in the experiment. Even fewer methods apply to general designs such as those that balance covariates of the treated and control groups, which are widely used precisely in the settings with small experimental samples and heterogeneous treatment effects (Harshaw et al., 2024).

In this paper, we propose a novel application of integer programming to identify the joint potential outcome distribution that maximizes the variance of the chosen estimator while being consistent with the randomization design and the observed marginal distributions of potential outcomes. Solving for this joint distribution *numerically* is what allows us to extend prior work (Aronow et al., 2014; Imbens & Menzel, 2021) which relies on well-known results of the optimality of the isotone copula for the difference-in-means estimator and completely randomized assignment. In partic-

[1]Google Research [2]University of Southern California [3]Meta (work completed while at Google Research). Correspondence to: Jean Pouget-Abadie <jeanpa@google.com>.

*Proceedings of the 42$^{nd}$ International Conference on Machine Learning*, Vancouver, Canada. PMLR 267, 2025. Copyright 2025 by the author(s).

ular, our method applies to all unconfounded (and known and probabilistic) assignment mechanisms, as well as any individualistic (and known and probabilistic) assignment mechanisms that verifies some bounded confoundedness. We examine conditionally unconfounded assignments as a special case. Furthermore, our method applies to any linear-in-treatment estimator, i.e. estimators of the form $\sum_i Z_i a_i + b_i$, and quadratic-in-treatment estimators, i.e. estimators of the form $\sum_i b_i + \sum_j Z_i Z_j a_{i,j}$, which include among others, difference-in-means, linear regression on the treatment variable and a covariate, and any doubly robust estimator fit out-of-sample. We provide asymptotic validity results and illustrate the application of our method to a simulated geographical experiment, using real data from the International Monetary Fund (IMF). Our method provides similar coverage and generally tighter confidence intervals across a variety of settings when compared to a widely used, but more conservative baseline.

In Section 2, we define and introduce our method for the illustrative setting of the completely randomized assignment and the difference-in-means estimator. In Section 3, we show that the same approach can be generalized to linear- and quadratic-in-treatment estimators. In Section 4, we show our method applies and is asymptotically valid for unconfounded assignments, conditionally unconfounded assignments, and individualistic with bounded confoundedness assignments. We conclude in Section 5 with practical simulations on real data.

### 1.1. Related works

The quantification of design uncertainty traces its roots to Neyman (1923), who studied the difference-in-means estimator under complete randomization in the finite population setting. Later work identified tight upper bounds for binary (Robins, 1988), interval (Stoye, 2010), and continuous (Aronow et al., 2014) outcomes. Imbens & Menzel (2021) extend the method of Aronow et al. (2014) with a resampling procedure they call the *causal bootstrap*, providing a more practical confidence interval construction.

Tighter variance estimates can be obtained when predictions are paired with informative covariates. Ding et al. (2019) and Wang et al. (2020) extend the work of Aronow et al. (2014) to the difference-in-means estimator in the presence of covariates, while Fan & Park (2010) propose the fewer-than-$n$ bootstrap method in the presence of covariates. Another body of literature studies design uncertainty for linear regression estimators. Freedman (2008), Schochet (2010), Samii & Aronow (2012), and Lin (2013) characterize the contexts in which classical sampling-based variance estimators are valid in the design setting, while Abadie et al. (2014) and Fogarty (2018) propose regression variance estimators specific to the finite-population setting. Abadie

et al. (2020) propose a novel variance estimator for linear regression under both design and sampling uncertainty that trades off between a classical ordinary least squares variance estimator and a novel design-based estimator based on the fraction of the population that is sampled.

Other studies expand design uncertainty estimation beyond Bernoulli or completely randomized designs. Dasgupta et al. (2015) and Lu (2016) study design uncertainty under factorial designs. Imai (2008) extends Neyman's upper bound to the matched-pairs design, which is a special case of covariate-balancing designs; Fogarty (2018) further extends this result to regression adjustment for matched pairs.

Neyman's original variance estimator is unbiased only for homogeneous ("strictly additive") treatment effects. Mukerjee et al. (2018) identify a less restrictive set of constraints on the potential outcomes and experimental design under which the variance of certain estimators is unbiasedly estimable. This approach relies on a "Q decomposition" of the variance, where the matrix Q must be derived for each experimental design. The approach fails for certain designs including matched pairs, where units in the same pair cannot be assigned the same treatment. Chattopadhyay & Imbens (2024) introduce two new variance decompositions for general designs and illustrate their application to completely randomized designs.

Like in (Aronow et al., 2014), our approach relies on identifying the variance-maximizing coupling of treated and control observations. We then apply the causal bootstrap method of Imbens & Rubin (2015) to generate confidence intervals. Our main contribution to these works is to generalize to new estimators and assignment mechanisms through the formulation of an integer program, which no longer requires the worst-case coupling to be known in closed-form. In a related optimization framework, Ji et al. (2023) show how to estimate bounds on a functional of the joint distribution via optimal transport. They do not explicitly solve for the optimal coupling, but their dual approach is robust to misspecification of marginal treated and control outcome probabilities in the presence of covariates. Finally, Harshaw et al. (2021) propose a minimal variance bound among all quadratic forms through a convex optimization procedure. Their method yields interesting admissibility results but only considers quadratic bounds. Our approach, inspired by the Frechet-Hoeffding bound approach used by Aronow et al. (2014) and Imbens & Menzel (2021), avoids this limitation.

## 2. Our causal bootstrap procedure

### 2.1. Notation and background

To build intuition, we briefly review Neyman's conservative bound and follow-up work on the difference-in-means estimator under a completely randomized assignment. We con-

sider a finite-population potential outcomes model in which each unit $i \in [N]$ is assigned to treatment ($Z_i = 0$) or control ($Z_i = 1$), and outcome $Y_i(Z_i)$ is observed. The average treatment effect is defined as $\tau = \frac{1}{N} \sum_{i \in [N]} Y_i(1) - Y_i(0)$. Let $N_1$ and $N_0$ be the number of treated and controlled units respectively. Define the difference in means estimator $\hat{\tau}_{\text{DIM}} = \frac{1}{N_1} \sum_{i:Z_i=1} Y_i - \frac{1}{N_0} \sum_{i:Z_i=0} Y_i$

and the *completely randomized design* that assigns exactly $N_1$ units to treatment and $N_0$ units to control. Neyman (1923) decomposed the variance of this estimator as

$$\mathbf{Var}_Z[\hat{\tau}_{\text{DIM}}] = \frac{S_1^2}{N_1} + \frac{S_0^2}{N_0} - \frac{S_\tau^2}{N}, \qquad (1)$$

where $S_z^2$ is the sample variances of the treated ($z = 1$) and control ($z = 0$) units, while $S_\tau^2$ is the sample variance of the unit-level treatment effects $Y_i(1) - Y_i(0)$. See Appendix A.8 for the full formulas. $S_\tau^2$ depends on the unknown joint distribution between treated and control outcomes since it contains terms $Y_i(1)Y_i(0)$, making it impossible to estimate consistently from experimental data in which only one of $Y_i(1)$ or $Y_i(0)$ is ever observed. While Neyman used the fact that variances are nonnegative to conservatively bound $S_\tau^2 \geq 0$, thus proposing an upper-bound of $\mathbf{Var}_Z[\hat{\tau}_{\text{DIM}}]$, this bound can be improved by identifying a joint distribution that maximizes $S_\tau^2$ while matching the observed marginal distributions of $Y(0)$ and $Y(1)$. Aronow et al. (2014) identify this variance-maximizing distribution to be the *isotone (assortative) copula*, which pairs the smallest treated outcome with the smallest control outcome, then the second-smallest, and so on. They derive analytical variance bounds using that joint distribution, while Imbens & Menzel (2021) propose a bootstrap-style approach, imputing the counterfactual outcomes using the isotone copula and resampling those imputed outcomes to generate bootstrap samples.

## 2.2. Our approach

In this work, we propose to identify the variance-maximizing joint distribution numerically using optimization instead of analytical derivations, allowing us to extend the least favorable copula approach to a broader class of estimators and assignments. Our goal is to find the joint distribution of potential outcomes $F_{0,1}$ which maximizes the variance of some estimator $\hat{\tau}$ under some assignment mechanism $Z$ and with some constraint $\mathcal{C}$ on the shape of this joint distribution.

$$\max_{Y_i(0), Y_i(1) \in \mathcal{Y}^2} \quad \mathbf{Var}_Z[\hat{\tau}] \qquad (2)$$
$$\text{s.t.} \quad F_{0,1} \in \mathcal{C}$$

There are multiple ways of constructing confidence intervals from the solution to this optimization objective. We propose to follow a similar procedure to the causal bootstrap

method proposed by Imbens & Menzel (2021): we impute unobserved potential outcomes using the least-favorable copula obtained above and drawing bootstrap samples via rerandomization of units to treatment. Confidence intervals are then constructed from the quantiles of this causal bootstrap distribution. We refer the reader to a full discussion in Appendix A.9.

We propose to solve the least-favorable copula problem numerically, by formulating the optimization problem in Eq. (2) in a solver-friendly way. To build intuition, we first consider the difference-in-means estimator. We will explore generalizations to a wider class of estimators in Section 3.1 and 3.2. The variance of the difference-in-means estimator has a quadratic closed-form for general assignments: $\mathbf{Var}_Z[\hat{\tau}] : \{Y_i(1), Y_i(0)\} \to \tilde{\mathbf{Y}}^T \mathbf{\Sigma}_{ZZ} \tilde{\mathbf{Y}}$, where $\tilde{\mathbf{Y}}$ is the vector of coordinates $\{N_1^{-1} Y_i(1) + N_0^{-1} Y_i(0)\}_{i=1...N}$, and $\mathbf{\Sigma}_{ZZ}$ is the covariance matrix of the random vector $\{Z_i\}_{i=1...N}$, with coordinates $(\mathbf{\Sigma}_{ZZ})_{ij} = \mathbf{Cov}[Z_i, Z_j]$. While it may be tempting to use this formulation directly, $\mathbf{\Sigma}_{ZZ}$ is positive semi-definite, making the objective straightforward to minimize, but more difficult to maximize under constraints, which is our goal (2).

Instead, we propose the following formulation: consider the indicator variables $X_{ik}^{(a)}$ with $a \in \{0, 1\}$ indexing the treatment assignment, $i \in [1, N]$ indexing the units in the population, and $y_k$ the $k^{th}$ outcome of $\mathcal{Y}$: $X_{ik}^{(a)} := \mathbb{I}(Y_i(a) = y_k)$. In particular, this notation implies that $\mathcal{Y}$ is discrete. For continuous outcomes, $\mathcal{Y}$ can be made discrete by replacing it with the support of observed outcomes. We provide asymptotic validity results of this approach in Section 4.2, though nothing prevents practitioners from expanding this discrete support of outcomes—beyond computational scalability, which we cover in Section 5—to cover other unseen outcomes. We can rewrite the expression of the variance of the difference-in-means estimator under general assignments as a quadratic form:

$$\mathbf{Var}_Z[\hat{\tau}] = \mathbf{X^T} \underbrace{\mathbf{Y^T} \mathbf{\Sigma}_{ZZ} \mathbf{Y}}_{\mathbf{Q}} \mathbf{X} = \mathbf{X^T Q X}, \qquad (3)$$

where $\mathbf{X} \in \{0, 1\}^{(2 \cdot N \cdot K)}$ and $\mathbf{Y} \in \mathbb{R}^{N \times (2N \cdot K)}$ are a vector and matrix representation of the indicator variables and potential outcomes respectively; cf. Appendix A.3 for details.

## 2.3. Adding constraints on the potential outcomes

An important part of our proposed algorithm is imposing realistic constraints on the missing potential outcomes that do not jeopardize the asymptotic validity of our bound while reducing the value of our objective (2), hence making our bound tighter. We constrain each unit $i$'s potential outcome $Y_i(a)$ to only match one possible value $y_k$, leading to the constraint $\sum_k X_{ik}^{(a)} = 1$. Additionally, because we observe

$N$ of the $2N$ potential outcomes, we set $X_{ik}^{(Z_i)} = 1$ if and only if $Y_i^{obs} = y_k$.

A clever observation by Aronow et al. (2014) is that the observed potential outcomes allow us to impose further constraints on the missing potential outcomes. While they focus entirely on the completely randomized assignment, which we use now to build intuition, we will generalize our results to other assignments in Section 4.1. If all units receive treatment with the same probability (different from 1 or 0), we expect the treated observed $F_1^{obs}$ and missing $F_1^{mis}$ marginals to match and the controlled observed $F_0^{obs}$ and missing $F_0^{mis}$ marginals to match. Formally, for any outcome $y$ and treatment assignment value $a \in \{0, 1\}$, consider

$$F_a^{obs}(y) := \frac{1}{N_a} \sum_{i=1}^{N} \mathbb{I}(Z_i = a, Y_i(a) = y),$$

$$F_a^{mis}(y) := \frac{1}{N_{1-a}} \sum_{i=1}^{N} \mathbb{I}(Z_i = 1 - a, Y_i(a) = y).$$

For a completely randomized assignment, each observed and missing marginal is equal in expectation to the true underlying marginal $F_a$. In other words, we have the following equality for any potential outcome $y$:

$$\mathbb{E}_Z[F_a^{mis}(y)] = \underbrace{\frac{1}{N} \sum_i \mathbb{I}(Y_i(a) = y)}_{F_a(y)} = \mathbb{E}_Z[F_a^{obs}(y)].$$

In our chosen parameterization, this equality on the marginal distributions can be written as a linear constraint on the variables $(X_{ik}^{(a)})$:

$$\forall a \in \{0, 1\}, \forall k, \ \sum_{i=1}^{N} \frac{Z_i}{N_1} X_{ik}^{(a)} - \frac{1 - Z_i}{N_0} X_{ik}^{(a)} = 0$$

### 2.4. Our proposed integer linear program

We now rewrite our objective in Eq. (2) as an integer program. For the completely randomized assignment and the difference-in-means estimator, we know from prior work that its solution is the isotone copula. The value of the integer program formulation will become clear as we generalize it beyond the completely randomized assignment, and beyond the difference-in-means estimator—settings for which closed-form solutions are not always known.

When indicated, the equations below are defined for all $a \in \{0, 1\}$, for all $i \in [1, N]$, and for all $y_k \in \mathcal{Y}$. As written, the optimization has binary variables and its constraints are all linear, but the objective is quadratic. Because the product of two binary variables is also a binary variable, the objective can be rewritten as a linear function of binary variables,

for which powerful solvers are available, using standard transformations; details are provided in Appendix A.3.

$$\max \ \mathbf{X^T Q X} \quad \text{for } \mathbf{X} \in \{0, 1\}^{N \cdot |\mathcal{Y}|} \quad (4)$$

$$\forall i, k, \quad X_{ik}^{(Z_i)} = 1 \text{ iff } Y_i^{obs} = y_k \quad (a)$$

$$\forall a, i, k, \quad X_{ik}^{(a)} = 0 \text{ iff } y_k \notin \text{supp}(F_a) \quad (b)$$

$$\forall a, i, \quad \sum_{k=1}^{K} X_{ik}^{(a)} = 1 \quad (c)$$

$$\forall a, k, \quad \sum_{i=1}^{N} X_{ik}^{(a)} \left( \frac{Z_i}{N_1} - \frac{1 - Z_i}{N_0} \right) = 0 \quad (d)$$

Constraint (4.b) is optional and allows us to account for practical assumptions we may have on the support of $F_a$. For example, we may be able to assume that $F_a \subset F_a^{obs}$, further reducing our estimate. Constraint (4.d) is not always feasible, when taken in conjunction with the other constraints on $(X_{ik}^{(a)})$. To ensure feasibility, we can relax these constraints for some chosen slackness parameter $\epsilon > 0$,

$$\forall a \in \{0, 1\}, \forall k, \ \left| \sum_{i=1}^{N} \frac{Z_i}{N_1} X_{ik}^{(a)} - \frac{1 - Z_i}{N_0} X_{ik}^{(a)} \right| \le \epsilon.$$

The regime of feasibility is captured in the following Lemma, a proof of which can be found in Appendix A.2.

**Lemma 2.1.** *For $\epsilon = 0$, there may be no solution to the constrained optimization below (4), unless $N_0 = N_1$. It admits a feasible solution for $\epsilon \ge 1/\min(N_0, N_1)$.*

As $\epsilon$ decreases, the worst case bound on the variance decreases, improving the performance of our estimator. We include further discussions on the choice of $\epsilon$ in Section 4.2.

## 3. Generalizing to new estimators

### 3.1. Linear-in-treatment estimators

So far, our results hold only for the difference-in-means estimator, which has been the primary object of study of previous work. In fact, they can be easily extended to a slightly more general class of estimators of the form: $\hat{\tau} := \sum_{i=1}^{N} Z_i a_i + b_i$, where $a_i$ and $b_i$ are constants with respect to the assignment mechanism and linear in the potential outcomes. We refer to such estimators as being *linear-in-treatment* (the second linearity assumption is then implied, for simplicity). Whether an estimator is linear-in-treatment depends on the assignment mechanism. Strictly speaking, the difference-in-means estimator is linear-in-treatment for a completely randomized assignment but not for a *Bernoulli assignment*, since the normalization constants $N_1^{-1}$ and $N_0^{-1}$ are not constants with respect to $Z$. The Horvitz-Thompson estimator (Horvitz & Thompson, 1952) is linear-

in-treatment for either assignment mechanism:

$$\hat{\tau} = \sum_i Z_i \underbrace{\left( \frac{Y_i(1)}{N \cdot P_i} + \frac{Y_i(0)}{N \cdot (1 - P_i)} \right)}_{a_i} + \underbrace{\frac{-Y_i(0)}{N \cdot (1 - P_i)}}_{b_i},$$

where $P_i = \mathbb{P}(Z_i = 1)$ is the probability that unit $i$ receives treatment. For any assignment mechanism and linear-in-treatment estimator, $\mathbf{Var}_Z[\hat{\tau}] = \sum_{i,j} a_i a_j \mathbf{Cov}[Z_i, Z_j] = \mathbf{X^T Q' X}$, where $\mathbf{Q'} := \mathbf{Y^T U^T \Sigma_{ZZ} U Y}$, with $\mathbf{Y}$ and $\mathbf{X}$ defined similarly to Section 2.4, and $\mathbf{U}$ such that $\mathbf{a} = \mathbf{U Y X}$, with $\mathbf{a}$ the vector of coordinates $\{a_i\}_{i \in [N]}$. The existence of $\mathbf{U}$ is given by the linearity of $a_i$ in the potential outcomes. The constraints in our integer program in Eq. 4 do not incorporate any estimator information, such that we can easily generalize our results to any linear-in-treatment estimator by replacing $\mathbf{Q}$ with $\mathbf{Q'}$.

### 3.2. Fitting regressions out-of-sample

In most practical applications, we also have access to covariates, which improve the precision of effect estimates when they are predictive of the outcomes. In this section and the next, we consider ways to extend our integer program to estimators that include covariate information.

One straightforward way to extend our previous analysis is to consider outcome-to-covariate functions that are fit out-of-sample. Consider the doubly-robust estimator as an illustrative example, where we replace the propensity scores by the true probabilities of each unit being treated, since we are motivated by randomized experiments where the probabilities are known. Let $\hat{f}_1$ and $\hat{f}_0$ be the *predicted* outcomes using functions that are fit out-of-sample and do not depend on $\mathbf{Z}$, and $\hat{\xi}_1 := Y_i - \hat{f}_1(W_i)$ and $\hat{\xi}_0 := Y_i - \hat{f}_1(W_i)$ the corresponding residuals, such that we write

$$\hat{\tau} := \left\{ \frac{1}{N} \sum_{i=1}^N \hat{f}_1(W_i) + \frac{Z_i \hat{\xi}_1}{P_i} - \hat{f}_0(W_i) - \frac{(1 - Z_i)\hat{\xi}_0}{1 - P_i} \right\}.$$

The variance of this estimator is equal to the variance of the Horvitz-Thompson estimator applied to the residuals $Y_i' \leftarrow Y_i - Z_i \hat{f}_1(W_i) - (1 - Z_i)\hat{f}_0(W_i)$. When taking the variance of $\hat{\tau}$ with respect to $\mathbf{Z}$ and replacing $Y_i' \leftarrow Y_i - Z_i \hat{f}_1(W_i) - (1 - Z_i)\hat{f}_0(W_i)$, we have:

$$\mathbf{Var_Z}[\hat{\tau}] = \mathbf{Var_Z} \left[ \frac{1}{N} \sum_{i=1}^N \frac{Z_i Y_i'}{P_i} - \frac{1}{N} \sum_{i=1}^N \frac{(1 - Z_i)Y_i'}{1 - P_i} \right]$$

We recognize the Horvitz-Thompson estimator applied to the residuals $Y_i'$. We can apply our method to $Y_i \leftarrow Y_i'$. In particular, the constraints on the marginal distributions now constrain the *residual* marginal distributions to match.

There are many practical settings in which fitted-out-of-sample functions are available. For example, such functions may be fitted on the same cohort before the start of the experiment. A typical and straightforward example is to use

a pre-treatment-period value of the outcome as a plug-in estimator (Abadie, 2005; Deng et al., 2013). If we expect growth of this variable, we can also account for its rate of growth and rescale the plug-in estimator with its expected value; however, fitting this growth rate using data from the experimental period depends on the treatment assignment $\mathbf{Z}$ and should be avoided. We include an example of this approach in Section 5. Another option is to rely on sample-splitting, where we split units into $C$ cohorts, chosen independently of the treatment assignment $\mathbf{Z}$. In that case, we can estimate a (pair of) function(s) per cohort using data from all other cohorts.

### 3.3. Quadratic-in-treatment estimators

Repeating a similar approach to Section 3.1 for estimators of the form $\hat{\tau} = \sum_i b_i + \sum_j Z_i Z_j a_{ij}$, we obtain the following quadratic form in the coefficients $a_{ij}$ when computing their variance: $\mathbf{Var}_Z[\hat{\tau}] = \sum_{i,j,k,l} a_{ij} a_{kl} \mathbf{Cov}[Z_i Z_j, Z_k Z_l]$. We can then repeat the approach of Section 3.1 as long as the coefficients $a_{ij}$ continue to be constant with respect to the treatment assignment and linear in the potential outcomes. In particular, we show in Appendix A.6 that a linear regression fitting observed outcome $Y_i^{obs}$ to the treatment variable $Z$ and a scalar covariate $X$ can be written in this form, with coefficients $a_{ij}$ that are linear in the potential outcomes. One advantage of this approach is that we are no longer required to do sample-splitting or use solely pre-period information to incorporate covariate information and reduce the size of our confidence intervals.

## 4. Generalizing to new mechanisms

We now show that our integer programming formulation in Eq. 4 can be generalized to a wider class of probabilistic assignment mechanisms for any of the estimators in Section 3.

### 4.1. Known and Probabilititistic assignments

Of the constraints in Eq. 4 requiring generalization, only (4.d) is restricted to assignments having $P_i = p \; \forall i$. In the more general setting of all *known* and *probabilistic* assignment, we no longer expect $F_a^{mis}$ and $F_a^{obs}$ to match exactly—or even approximately. If the treatment probabilities are known however, we can construct an unbiased estimator of the true marginal $F_a$ from both observed and missing potential outcomes. For $a \in \{0, 1\}$ and $y_k \in \mathcal{Y}$, consider the marginal estimators:

$$\hat{F}_a^{obs}(y_k) := \frac{1}{N} \sum_i \frac{Z_i^a (1 - Z_i)^{1-a}}{P_i^a (1 - P_i)^{1-a}} X_{ik}^{(a)} \quad \text{and}$$

$$\hat{F}_a^{mis}(y_k) := \frac{1}{N} \sum_i \frac{Z_i^{1-a}(1 - Z_i)^a}{P_i^{1-a}(1 - P_i)^a} X_{ik}^{(a)}$$

Through iterated expectations, it holds that $\mathbf{E}[\hat{F}_a^{obs}(y_k)] = \mathbf{E}[\hat{F}_a^{mis}(y_k)]$. We can thus reformulate constraints (d) as follows, for all $a, k, b \in \{0,1\} \times |\mathcal{Y}| \times \{0,1\}$:

$$(-1)^b \sum_{i=1}^N X_{ik}^{(a)} \left( \frac{Z_i}{P_i} - \frac{1 - Z_i}{1 - P_i} \right) \leq \epsilon N \,.$$

The resulting integer program applies to any known, and probabilistic assignment mechanism.

## 4.2. Asymptotic Validity

Just because we can formulate (and solve) an integer program for any known and probabilistic assignment mechanism, does not mean our bound is asymptotically valid. Our procedure is *guaranteed* to upper-bound the true variance only if the true joint distribution is also a feasible solution of our integer program. This could be violated if (a) the discrete support we choose, equal to the support of observed outcomes, does not also encompass the support of the missing potential outcomes, or (b) the realized distributions $F_a^{mis}$ and $F_a^{obs}$ do not match exactly. Recall that we proposed relaxing constraint (4.d) to hold with $\epsilon$ slackness to ensure feasibility of our integer program. In fact, by bounding the probability that the observed and missing marginal distributions are uniformly within an $\epsilon$ distance of each other, we can provide a rate for the probability that our procedure upper-bounds the true variance as $N$ grows:

**Theorem 4.1.** *For any probabilistic unconfounded assignment mechanism,* $\mathbb{P}(V^* \geq \mathbf{Var}_Z[\hat{\tau}]) \geq 1 - \beta$, *where $V^*$ is the output of our integer program,* $\beta := 8 \exp\left(-\frac{\epsilon^2}{4} N \tilde{P}\right) + \frac{32}{N^2 \tilde{P}^2} \sum_{i,j \in [N]} \mathbf{Cov}(Z_i, Z_j)$, $\epsilon$ *is the slackness parameter in* (4.d), $\tilde{P} = \min(P, 1 - P)$, *and* $P = N^{-1} \sum_i P_i$.

As expected, as the sample size $N$ and the slackness $\epsilon$ that we choose grows, the probability of upper-bounding the true variance increases. We can therefore justify our approach theoretically for any *unconfounded* (and known and probabilistic) assignments. We will examine the special case of conditionally unconfounded assignments in Section 4.3. We can further refine the result above if the assignment mechanism is also individualistic:

**Corollary 4.2.** *For any probabilistic, unconfounded, and individualistic assignment mechanism, Theorem 4.1 holds with* $\beta = 8 \exp\left(-\frac{\epsilon^2}{4} N \tilde{P}\right) + 8 \exp\left(-\frac{1}{2} N \tilde{P}^2\right)$.

What of confounded assignments? In fact, we can also prove our approach is asymptotically valid if the assignment is individualistic with bounded confoundedness, as defined by (Tan, 2006; De Bartolomeis et al., 2024). Adopting similar notation to Theorem 4.1,

**Theorem 4.3.** *Suppose that* $(Y_i(0), Y_i(1), Z_i)$ *are i.i.d*

across $i \in [n]$, *and suppose that*

$$\left| \frac{\mathbb{P}(Z = a|Y(0), Y(1))}{\mathbb{P}(Z = a)} - 1 \right| \leq \delta \,,$$

*for some $\delta > 0$, such that $\delta$ controls the effect of confoundedness, with $\delta = 0$ corresponding to unconfoundedness. If the slackness satisfies $\epsilon \geq \delta$, then*

$$\mathbb{P}\left(V^* \geq \mathbf{Var}_Z[\hat{\tau}]\right) \geq 1 - \beta \,,$$

*where* $\beta := 1 - 8e^{-2N\epsilon_0^2} - 8e^{-N\tilde{P}\epsilon_1^2/3}$, $\epsilon_0 = \frac{(\epsilon - \delta)\tilde{P}(1+\delta)}{2 + \delta + \epsilon}$, *and* $\epsilon_1 = \frac{\epsilon - \delta}{2 + \delta + \epsilon}$.

Details can be found in Appendix A.5. We leave the generalization to general confounded assignments to future work.

## 4.3. Conditionally unconfounded assignments

It is possible for the imbalance in $F_a^{mis}$ and $F_a^{obs}$ to be fully captured by an observed covariate $W_i$, as is the case for conditionally unconfounded assignments, which verify $Z_i \perp Y_i(1), Y_i(0)|W_i$. In that case, we have at least two options for formulating the integer program. The first option, and the most straightforward one, is to re-use the integer program from Section 4.1 with $P_i = \mathbb{P}(Z_i|W_i = W_i)$. While our previous results follow through trivially, this approach does not leverage all the covariate information available.

A second approach that leverages the covariate information is to impose equality of the *conditional* marginal distributions, reweighted by the suitable inverse probabilities. In other words, for all $a \in \{0,1\}, y_k \in \mathcal{Y}$ and values $w$ of covariate $W$, consider the conditional marginal estimators:

$$\hat{F}_a^{obs}(y_k|w) := \sum_{i:W_i=w} \frac{Z_i^a(1 - Z_i)^{1-a}}{P_i(w)^a(1 - P_i(w))^{1-a}} \frac{X_{ik}^{(a)}}{|\{w\}|}$$

$$\hat{F}_a^{mis}(y_k|w) := \sum_{i:W_i=w} \frac{Z_i^{1-a}(1 - Z_i)^a}{P_i(w)^{1-a}(1 - P_i(w))^a} \frac{X_{ik}^{(a)}}{|\{w\}|}$$

Here, $P_i(w) := \mathbb{P}(Z_i = 1|W_i = w)$ is the probability that unit $i$ receives treatment $Z_i = a$ given covariate $W_i = w$, and $|\{w\}|$ denotes the number of units with covariate $W_i = w$. As a result, the following equalities hold for all $(a, y_k, w)$ triplets: $\mathbb{E}_Z\left[\hat{F}_a^{obs}(y_k|w)\right] = \mathbb{E}_Z\left[\hat{F}_a^{mis}(y_k|w)\right]$.

These equalities can be expressed in a solver-friendly way, similar to the formulation of the (4.d) constraints in Section 4.1. Doing so allows us to incorporate covariate information into our variance estimator for tighter confidence intervals, since it adds additional constraints to the optimization problem. On the other hand, the empirical conditional marginal distribution $\hat{F}_a(y_k|w)$ converges to its expectation more slowly than the unconditional marginal, which will negatively affect the performance of our variance bound in small samples. This is especially true when $W$ is high-dimensional or continuous.

# 5. Simulations on real data

## 5.1. Empirical Results

Our approach is well-motivated for geographical experiments, which assign treatment and control to, e.g. countries or Designated Marketing Areas. This is due to the fact that geographical regions are fixed in number, meaning design uncertainty likely dominates sampling uncertainty. Additionally, there are often few available regions (on the order of hundreds) with significant heterogeneity in potential outcomes. As a result, (a) sample uncertainty quantification methods are poor approximations of the design uncertainty of our estimator, and (b) practitioners rely on non-standard designs and estimators to achieve statistical power. This is precisely the setting in which existing design uncertainty quantification methods fall short.

**Dataset**  To work with a realistic dataset, we look to the International Monetary Fund's publicly-available Gross Domestic Product (GDP) country-level report for the years 2017–2019, restricted to the top 50 countries by GDP (IMF, 2024). While a coordinated worldwide randomized experiment affecting GDP is implausible, we can make the more reasonable assumption that some global company's metric-of-interest in each country is proportional to that country's GDP, such that the simulation results would approximately hold if such a company were to run a country-level experiment. Our outcome of interest is GDP in 2019, with 2017–2018 GDP acting as pre-period covariates.

**Settings**  To showcase the validity and usefulness of our method, we compare two assignment strategies: the standard complete randomization design ("Compl. R.") treating exactly half of the countries, and a matched-pairs design ("Matched Pairs"). The matched-pairs design is constructed by matching the country with the largest GDP in 2018 with the country with the second largest GDP for that year, the third largest with the fourth largest, and so on. We consider two estimators: first, the standard "no-covariates" difference-in-means estimator, and second, the doubly-robust estimator from Section 3.2, using pre-period covariates from 2018 GDP adjusted for growth using the historical change between 2017 and 2018, as described in Section 3.2.

**Baselines**  Finally, we compare our method to three baselines. Difference-in-means (and its doubly-robust variant) admits conservative, analytical bounds for its sampling variance under both complete randomization and matched-pairs; see Appendix A.7. We form symmetric, Neyman-style confidence intervals based on these bounds. We next consider the standard sampling bootstrap, which samples with replacement $N$ units $1,000$ times, and reports the empirical confidence intervals as quantiles of the resulting estima-

tor distribution. We also compare to the causal bootstrap method suggested by (Aronow et al., 2014; Imbens & Menzel, 2021), which resamples from the isotone copula. This method is only theoretically justified in the complete randomization setting for the difference-in-means estimator. All reported confidence intervals have $95\%$ nominal coverage. We also report the "true confidence intervals" as defined by the $2.5\%$ and $97.5\%$ quantiles of the distribution of the estimate when resampling the assignment according to each design 500 times.

**Analysis**  Select results are summarized in Tables 1–2. A more complete set of results can be found in Appendix A.1, including implementation details and a reference to the open-source solver. For any experiment with complete randomization, our optimal causal bootstrap mechanism and the isotone copula bootstrap yield the same results down to each simulation as expected: our optimal causal bootstrap recovers the well-known variance-maximizing joint distribution exactly. Across all three inference methods and for the ground truth confidence intervals (CIs), the CIs are greatly reduced by incorporating pre-period information. These CIs are further reduced by the matched-pairs design, with more discussion in Appendix A.1

All three bootstrap methods achieve lower-than-nominal coverage for the completely randomized design, due to the heavy-tailed nature of GDP data, which causes $\hat{F}_a$ to converge slowly. This problem is particularly acute for the completely randomized design, which has high likelihood of assigning the largest GDP countries to the same treatment group. By contrast, the matched-pairs design balances the treated and controlled outcomes, improving the convergence of $\hat{F}_a$ and thus the coverage of the sampling bootstrap and our method. We provide more discussion of the observed undercoverage in Section A.10.

The conservative variance estimators fare well at achieving close-to-nominal coverage. For the doubly robust estimator under matched-pairs—the setup which achieves the best power here—the conservative estimator has the narrowest CI widths for small effect sizes (less than 2.5%), but for large effect size (over 10%), its CI widths become larger than those of the sampling bootstrap. Our optimal causal bootstrap maintains a stable of improvement of at least 11% in CI widths relative to the sampling bootstrap, across effects sizes. We emphasize that analytical variance bounds are not typically available for more complex designs, whereas our procedure extends as long as the treatment covariance matrix is available.

For the matched-pairs design, the isotone copula bootstrap is no longer known to be a conservative estimator. In practice, we find its resulting CIs are unrealistically narrow: in Table 1, they never intersect with zero, leading to a coverage of 0%. This is not surprising in a well-matched A/A setting

| Covariates | True CI | | Inference Method | Coverage | | Median CI | |
|---|---|---|---|---|---|---|---|
| | Compl. R. | Matched Pairs | | Compl. R. | Matched Pairs | Compl. R. | Matched Pairs |
| None | 3574 | 762 | Sampling Boot. | 90% | 100% | 3967 | 4110 |
| | | | Conservative Var. | 98% | 100% | 3991 | 1152 |
| | | | Isotone Copula | 87% | 0% | 3348 | 44 |
| | | | Opt. Causal Boot. | 87% | 100% | 3348 | 2233 |
| 2018 GDP | 142 | 46 | Sampling Boot. | 91% | 100% | 141 | 142 |
| | | | Conservative Var. | 94% | 95% | 143 | 49 |
| | | | Isotone Copula | 90% | 91% | 133 | 41 |
| | | | Opt. Causal Boot. | 90% | 100% | 133 | 102 |

Table 1: "True CI" and "Median CI" correspond to $95\%$ CI *widths* in an A/A test under two designs and two estimators.

| Covariates | Effect | True CI | | Inference Method | Median CI | | Power | |
|---|---|---|---|---|---|---|---|---|
| | | Compl. R. | Matched Pairs | | Compl. R. | Matched Pairs | Compl. R. | Matched Pairs |
| None | 2.5% | 3618 | 772 | Sampling Boot. | 4015 | 4160 | 0.10 | 0.00 |
| | | | | Conservative Var. | 4040 | 1215 | 0.02 | 0.00 |
| | | | | Isotone Copula | 3395 | – | 0.13 | – |
| | | | | Opt. Causal Boot. | 3395 | 2256 | 0.13 | 0.00 |
| | 10% | 3752 | 800 | Sampling Boot. | 4181 | 4306 | 0.10 | 0.00 |
| | | | | Conservative Var. | 4149 | 1464 | 0.02 | 0.00 |
| | | | | Isotone Copula | 3540 | – | 0.13 | – |
| | | | | Opt. Causal Boot. | 3540 | 2341 | 0.13 | 0.00 |
| 2018 GDP | 2.5% | 101 | 45 | Sampling Boot. | 111 | 109 | 0.32 | 0.20 |
| | | | | Conservative Var. | 117 | 84 | 0.24 | 0.46 |
| | | | | Isotone Copula | 99 | – | 0.39 | – |
| | | | | Opt. Causal Boot. | 99 | 97 | 0.39 | 0.25 |
| | 10% | 90 | 64 | Sampling Boot. | 185 | 235 | 1.00 | 1.00 |
| | | | | Conservative Var. | 189 | 310 | 1.00 | 0.93 |
| | | | | Isotone Copula | 154 | – | 1.00 | – |
| | | | | Opt. Causal Boot. | 154 | 203 | 1.00 | 1.00 |

Table 2: "True CI" and "Median CI" correspond to $95\%$ CI *widths* for multiplicative effects under two designs and two estimators.

where the difference of outcomes across pairs is larger than the difference within pairs. In that case, the isotone copula imputes the counterfactual outcome of each unit with the observed outcome of its paired unit. No matter how many times one samples from this imputed copula, the estimated effect within each pair will be the same, leading to an estimated variance of zero. However, the point estimate of our estimator can be non-zero, even for a well-matched A/A test, if the paired units are not exactly identical, leading to zero coverage. As a result, we do not report the median CIs of the isotone copula bootstrap in Table 2 for the matched-pairs design. On the other hand, both the sampling bootstrap and our optimal causal bootstrap have satisfactory coverage in both complete randomization and matched pair designs. Across all experiments, our method reduces CI widths by at least $10\%$ compared to the naive bootstrap, with no loss in power or coverage, with an exceptional 50% reduction in CI widths for the difference-in-means estimator with injected multiplicative effects.

## 5.2. Scalability Analysis

As mentioned, we focus on the top 50 countries by GDP for our simulation analysis. For a dataset of this size, our integer program took 671 seconds on average for the matched pairs design, and 1057 seconds on average for complete randomization. Matched pairs is faster because of the sparsity of its covariance matrix. However, complete randomization also admits a linear programming formulation for the worst-case copula, which took only 22 seconds on average to solve. We used this faster implementation in our experiments for the sake of expedience.

To assess the scalability of our formulation, we solved for the worst-case copula under a null effect, varying the number of units. Figures 1–2 presents the results of our scaling analysis, where each data point in the plot represents an average over 10 simulation runs, along with $95\%$ confidence intervals based on standard errors.

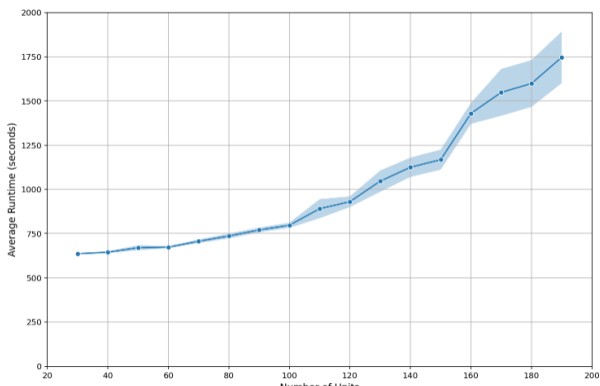

Figure 1: Runtime scaling for matched pairs, integer program formulation.

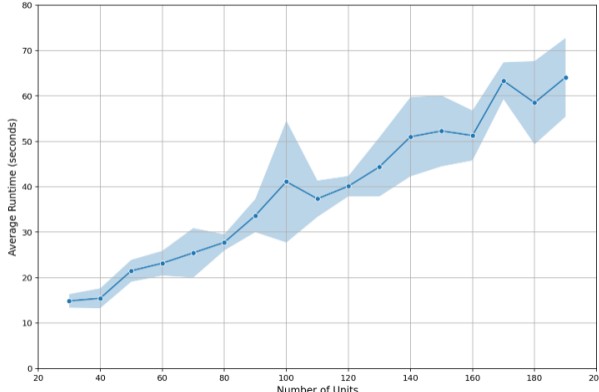

Figure 2: Runtime scaling for complete randomization, linear program formulation.

## 6. Conclusion

We propose an integer program approach to design-uncertainty quantification for linear- and quadratic-in-treatment estimators for known and probabilistic assignments, that are unconfounded, conditionally unconfounded, or individualistic with bounded confoundedness. Future work could look at new classes of estimators or provide asymptotic validity results for general confounded mechanisms. There is also the possibility of *unknown* probabilistic conditionally unconfounded assignmentsm, in which case the probabilities $P_i$ would be replaced with estimated propensity scores. The validity of our approach would then depend on the quality of these estimates. The limitations of our work are around computational scaling, which is limited to—but also best motivated for—small datasets with hundreds of units.

## Impact Statement

This paper presents work whose goal is to advance the field of Machine Learning. There are many potential societal consequences of our work, none of which we feel must be specifically highlighted here.

## Acknowledgments

Adel Javanmard is supported in part by the NSF Award DMS-2311024, the Sloan fellowship in Mathematics, an Adobe Faculty Research Award, an Amazon Faculty Research Award, and an iORB grant form USC Marshall School of Business. The authors are grateful to anonymous reviewers for their feedback on improving this paper.

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

# A. Appendix

## A.1. Complete Simulation Results

Integer programs to compute the optimal causal bootstrap imputation were solved using the CP-SAT solver (https://developers.google.com/optimization/cp/cp_solver). Experiments were parallelized on a cloud CPU cluster using 100 workers. Runtime for the final experiments was approximately 2 hours. Smaller runs of approximately 10 minutes were used to debug the simulation code. Tables 4 and 3 report on the complete simulation results on CI width and power for multiplicative and additive effects, respectively. Table 1 in the main body contains complete results for A/A tests.

| Covariates | Effect Size | Inference Method | Median CI | | Power | |
|---|---|---|---|---|---|---|
| | | | Compl. R. | Matched Pairs | Compl. R. | Matched Pairs |
| None | 82 | Sampling Boot. | 3967 | 4110 | 0.10 | 0.00 |
| None | 82 | Conservative Var. | 3991 | 1152 | 0.02 | 0.00 |
| None | 82 | Isotone Copula | 3348 | – | 0.13 | – |
| None | 82 | Opt. Causal Boot. | 3348 | 2230 | 0.13 | 0.00 |
| None | 164 | Sampling Boot. | 3967 | 4110 | 0.10 | 0.00 |
| None | 164 | Conservative Var. | 3991 | 1152 | 0.03 | 0.00 |
| None | 164 | Isotone Copula | 3348 | – | 0.14 | – |
| None | 164 | Opt. Causal Boot. | 3348 | 2230 | 0.14 | 0.00 |
| 2018 GDP | 82 | Sampling Boot. | 141 | 142 | 0.61 | 0.78 |
| 2018 GDP | 82 | Conservative Var. | 143 | 49 | 0.61 | 1.00 |
| 2018 GDP | 82 | Isotone Copula | 133 | – | 0.68 | – |
| 2018 GDP | 82 | Opt. Causal Boot. | 133 | 102 | 0.68 | 1.00 |
| 2018 GDP | 164 | Sampling Boot. | 141 | 142 | 1.00 | 1.00 |
| 2018 GDP | 164 | Conservative Var. | 143 | 49 | 1.00 | 1.00 |
| 2018 GDP | 164 | Isotone Copula | 133 | – | 1.00 | – |
| 2018 GDP | 164 | Opt. Causal Boot. | 133 | 101 | 1.00 | 1.00 |

Table 3: Complete results for $95\%$ CI widths and power for injected **additive** effects.

As a footnote, we observe that the "true CI" widths increase with effect size for both designs when there are no covariates, as expected. However, they sometimes decrease with effect size with covariates. This is due to the fact that the covariates model some growth in outcomes, and these models are not fit in-sample. Therefore, it is possible for variance reduction to be largest for nonzero effect sizes.

| Covariates | Effect Size | Inference Method | Median CI | | Power | |
|---|---|---|---|---|---|---|
| | | | Compl. R. | Matched Pairs | Compl. R. | Matched Pairs |
| None | 2.5% | Sampling Boot. | 4015 | 4160 | 0.10 | 0.00 |
| None | 2.5% | Conservative Var. | 4040 | 1215 | 0.02 | 0.00 |
| None | 2.5% | Isotone Copula | 3395 | – | 0.13 | – |
| None | 2.5% | Opt. Causal Boot. | 3395 | 2256 | 0.13 | 0.00 |
| None | 5.0% | Sampling Boot. | 4068 | 4206 | 0.10 | 0.00 |
| None | 5.0% | Conservative Var. | 4079 | 1298 | 0.01 | 0.00 |
| None | 5.0% | Isotone Copula | 3447 | – | 0.13 | – |
| None | 5.0% | Opt. Causal Boot. | 3447 | 2284 | 0.13 | 0.00 |
| None | 7.5% | Sampling Boot. | 4121 | 4260 | 0.10 | 0.00 |
| None | 7.5% | Conservative Var. | 4114 | 1381 | 0.02 | 0.00 |
| None | 7.5% | Isotone Copula | 3495 | – | 0.13 | – |
| None | 7.5% | Opt. Causal Boot. | 3495 | 2314 | 0.13 | 0.00 |
| None | 10.0% | Sampling Boot. | 4181 | 4306 | 0.10 | 0.00 |
| None | 10.0% | Conservative Var. | 4149 | 1464 | 0.02 | 0.00 |
| None | 10.0% | Isotone Copula | 3540 | – | 0.13 | – |
| None | 10.0% | Opt. Causal Boot. | 3540 | 2341 | 0.13 | 0.00 |
| 2018 GDP | 2.5% | Sampling Boot. | 111 | 109 | 0.32 | 0.20 |
| 2018 GDP | 2.5% | Conservative Var. | 117 | 84 | 0.24 | 0.46 |
| 2018 GDP | 2.5% | Isotone Copula | 99 | – | 0.39 | – |
| 2018 GDP | 2.5% | Opt. Causal Boot. | 99 | 97 | 0.39 | 0.25 |
| 2018 GDP | 5.0% | Sampling Boot. | 122 | 121 | 0.93 | 0.98 |
| 2018 GDP | 5.0% | Conservative Var. | 124 | 148 | 0.89 | 0.75 |
| 2018 GDP | 5.0% | Isotone Copula | 102 | – | 0.97 | – |
| 2018 GDP | 5.0% | Opt. Causal Boot. | 102 | 126 | 0.97 | 0.92 |
| 2018 GDP | 7.5% | Sampling Boot. | 154 | 169 | 1.00 | 1.00 |
| 2018 GDP | 7.5% | Conservative Var. | 152 | 228 | 1.00 | 0.88 |
| 2018 GDP | 7.5% | Isotone Copula | 126 | – | 1.00 | – |
| 2018 GDP | 7.5% | Opt. Causal Boot. | 126 | 162 | 1.00 | 1.00 |
| 2018 GDP | 10.0% | Sampling Boot. | 185 | 235 | 1.00 | 1.00 |
| 2018 GDP | 10.0% | Conservative Var. | 189 | 310 | 1.00 | 0.93 |
| 2018 GDP | 10.0% | Isotone Copula | 154 | – | 1.00 | – |
| 2018 GDP | 10.0% | Opt. Causal Boot. | 154 | 203 | 1.00 | 1.00 |

Table 4: Complete results for 95% CI widths and power for injected **multiplicative** effects.

### A.2. Proof of Lemma 2.1 in Section 2.3

To show the first part of the Lemma 2.1, we observe that for any outcome $y_k$, setting $\epsilon = 0$ implies $F_a^{obs}(y_k) = F_a^{mis}(y_k)$. The observed distribution $F_a^{obs}(y_k)$ is already determined and not in our control. Further, $N_0 F_1^{obs}(y_k) = N_0 F_1^{mis}(y_k) = \sum_{i=1}^{N}(1-Z_i)X_{ik}^{(1)}$ which is an integer value. Since $F_1^{obs}(y_k)$ is already given, it may be that there exists an outcome $y_k$ such that $N_0 F_1^{obs}(y_k)$ is not an integer, in which case the $(d)$ constraints above are not feasible when $\epsilon = 0$.

We now tackle the second statement of Lemma 2.1. The constraints corresponding to $a = 1$ are decoupled from those corresponding to $a = 0$. Here, we focus on $a = 1$ and the case of $a = 0$ can be treated similarly. Define the shorthand

$$p_k := F_1^{obs}(y_k) = \sum_{i=1}^{N} X_{ik}^{(1)} \frac{Z_i}{N_1} \,,$$

for every $y_k \in \mathrm{supp}(F_1^{obs})$. Note that the values of $p_k$ are given by constraint (a) and $\sum_k p_k = 1$. Let $m := |\mathrm{supp}(F_1^{obs})|$ and $C$ be the set of controlled units ($|C| = N_0$). For each $k \in [m]$, we let $A_k \subset C$ be an arbitrary set with $|A_k| = \lfloor N_0 p_k \rfloor$, such that the sets $A_k$ are disjoint. Note that

$$| \cup_{k=1}^{m} A_k | = \sum_{k=1}^{m} |A_k| = \sum_{k=1}^{m} \lfloor N_0 p_k \rfloor \leq N_0 \sum_{i=1}^{m} p_k = N_0 \,.$$

Therefore, the number of controlled units not covered by any of $A_k$ is at most

$$r := |C \backslash \cup_{i=1}^{k} A_k| = N_0 - \sum_{k=1}^{m} \lfloor N_0 p_k \rfloor = \sum_{k=1}^{m} (N_0 p_k - \lfloor N_0 p_k \rfloor) \leq \sum_{k=1}^{m} 1 = m \,.$$

We next increase the size of $|A_1|, \ldots, |A_r|$ by adding one element of $C \backslash \cup_{i=1}^{k} A_k$ to each of them (Note that this is possible since as we showed above $r \leq m$). This way, we have

$$|A_k| = \lfloor N_0 p_k \rfloor + 1, \quad \text{for } k = 1, \ldots, r,$$
$$|A_k| = \lfloor N_0 p_k \rfloor, \quad \text{for } k = r+1, \ldots, m \,,$$

the sets $A_k$ are disjoint and their union covers $C$.

We are now ready to construct a feasible solution as follows. For $i \in C$ (i.e., a controlled unit), we set

$$X_{ik}^{(1)} = \begin{cases} 1 & i \in A_k, \\ 0 & \text{otherwise.} \end{cases}$$

In addition, for $i \in C^c$ (i.e., a treated unit), we set

$$X_{ik}^{(1)} = 1 \quad \text{iff} \quad Y_i(1) = y_k \,. \tag{5}$$

Constraints (a) and (c) are clearly satisfied as $X_{ik}^{(1)}$ are binary variables that are zero outside $\mathrm{supp}(F_1^{obs})$. Condition $(a)$ is also satisfied by (5) and noting that for $i \in C^c$, we have $Y_i^{obs} = Y_i(1)$. To verify constraint (c), note that if $i \in C$, only one of $X_{ik}^{(1)}$ is one and the rest are zero because the sets $A_k$ are disjoint. The same holds for $i \in C^c$ clearly by the construction in (5).

We next verify constraint (d). We have

$$\sum_{i=1}^{N} X_{ik}^{(1)} \frac{Z_i}{N_1} - \sum_{i=1}^{N} X_{ik}^{(1)} \frac{1-Z_i}{N_0} = p_k - \sum_{i \in C} \frac{1}{N_0} X_{ik}^{(1)} = p_k - \frac{|A_k|}{N_0} \,.$$

For $k = 1, \ldots, r$, we have

$$\left| p_k - \frac{|A_k|}{N_0} \right| = \left| p_k - \frac{\lfloor N_0 p_k \rfloor + 1}{N_0} \right| = \left| \frac{1 - (N_0 p_k - \lfloor N_0 p_k \rfloor)}{N_0} \right| \leq \frac{1}{N_0} \,.$$

Likewise, for $k = r+1, \ldots, m$ we have

$$\left| p_k - \frac{|A_k|}{N_0} \right| = \left| p_k - \frac{\lfloor N_0 p_k \rfloor}{N_0} \right| = \left| \frac{N_0 p_k - \lfloor N_0 p_k \rfloor}{N_0} \right| \leq \frac{1}{N_0}.$$

Combining the previous two inequalities, we obtain that constraint (d) is satisfied for $\epsilon \geq 1/N_0$ for $a = 1$.

Following the same procedure for $a = 0$, we get that there exists a feasible solution to optimization (4) for $\epsilon \geq 1/\min(N_0, N_1)$.

When $N_0 = N_1$, note that in this case we can form a one-to-one matching of units in the control set to units in the treatment set. Take any such matching, and for $i \in N$ let $k_i$ be the outcome of the unit that $i$ is matched to. Set $X_{ik_i}^{(1-Z_i)} = 1$ and $X_{ik}^{(1-Z_i)} = 0$ for all $k \neq k_i$. Set $X_{ik}^{(Z_i)} = 1$ if $Y_i^{obs} = y_k$ and $X_{ik}^{(Z_i)} = 0$ otherwise. By construction, this solution satisfies constraints (4).a–(4).c and also constraints (4).d for $\epsilon = 0$.

### A.3. Constructing the Integer Linear Program in Section 2.4

Let $\mathbf{y}^{(\mathbf{a})} \in \mathbb{R}^{|\mathcal{Y}|}$ be the vector of potential outcomes $\mathcal{Y}$. Consider the following matrix and vector variables, with $\mathbf{Y}^{(\mathbf{a})}$ an $N \times (N \cdot K)$ matrix and $\mathbf{X}^{(\mathbf{a})}$ an $(N \cdot K)$ vector:

$$\mathbf{Y}^{(\mathbf{a})} := \begin{pmatrix} \mathbf{y}^{(\mathbf{a})\mathbf{T}} & 0 & \cdots & 0 \\ 0 & \mathbf{y}^{(\mathbf{a})\mathbf{T}} & 0 & \vdots \\ \vdots & 0 & \ddots & 0 \\ 0 & \cdots & 0 & \mathbf{y}^{(\mathbf{a})\mathbf{T}} \end{pmatrix}$$

$$\mathbf{X}^{(\mathbf{a})} = \begin{pmatrix} X_{11}^{(a)} & \cdots & X_{ik}^{(a)} & X_{i(k+1)}^{(a)} & \cdots & X_{NK}^{(a)} \end{pmatrix}^T$$

We further consider the following combined matrices:

$$\mathbf{Y} := \left( \tfrac{1}{N_0} \mathbf{Y}^{(\mathbf{0})}, \tfrac{1}{N_1} \mathbf{Y}^{(\mathbf{1})} \right) \quad \mathbf{X} := \begin{pmatrix} \mathbf{X}^{(\mathbf{0})} \\ \mathbf{X}^{(\mathbf{1})} \end{pmatrix}$$

While the notation is clumsy, it allows us to rewrite $\mathbf{Var}_Z[\hat{\tau}]$ in a solver-friendly way, with $\mathbf{X}$ a vector and our objective the quadratic function. Recall that the variance of the difference-in-means estimator has a quadratic closed-form for general assignments: $\mathbf{Var}_Z[\hat{\tau}] : \{Y_i(1), Y_i(0)\} \rightarrow \hat{\mathbf{Y}}^T \Sigma_{ZZ} \hat{\mathbf{Y}}$, where $\hat{\mathbf{Y}}$ is the vector of coordinates $\{N_1^{-1} Y_i(1) + N_0^{-1} Y_i(0)\}_{i=1\ldots N}$, and $\Sigma_{ZZ}$ is the covariance matrix of the random vector $\{Z_i\}_{i=1\ldots N}$, with coordinates $(\Sigma_{ZZ})_{ij} = \mathbf{Cov}[Z_i, Z_j]$. Rewriting $\mathbf{Var}_Z[\hat{\tau}]$ in terms of $\mathbf{X}$ and $\mathbf{Y}$, we obtain:

$$\mathbf{Var}_Z[\hat{\tau}] = \mathbf{X}^{\mathbf{T}} \underbrace{\mathbf{Y}^{\mathbf{T}} \Sigma_{ZZ} \mathbf{Y}}_{\mathbf{Q}} \mathbf{X} = \mathbf{X}^{\mathbf{T}} \mathbf{Q} \mathbf{X}.$$

Because $\Sigma_{ZZ}$ is positive definite for any probabilistic assignment, it can be decomposed into the product of a matrix and its transpose $\mathbf{R}^{\mathbf{T}} \mathbf{R}$ by the Cholesky decomposition. Thus, $\mathbf{Q}$ can be written as the positive semi-definite matrix $\mathbf{Q} = \mathbf{Y}^{\mathbf{T}} \mathbf{R}^{\mathbf{T}} \mathbf{R} \mathbf{Y}$.

Furthermore, in the objective (4) we have products of binary variables of the form $X_{ik}^{(a)} \cdot X_{j\ell}^{(b)}$. We replace this product with a new binary variable $X_{ik,j\ell}^{(a,b)} \in \{0,1\}$, and introduce the following linear constraints to ensure the variable corresponds to the product of the original variables:

$$X_{ik,j\ell}^{(a,b)} \leq X_{ik}^{(a)}$$
$$X_{ik,j\ell}^{(a,b)} \leq X_{j\ell}^{(b)}$$
$$X_{ik,j\ell}^{(a,b)} \geq X_{ik}^{(a)} + X_{j\ell}^{(b)} - 1$$

It is easy to check that $X_{ik,j\ell}^{(a,b)} = 1$ if and only if both $X_{ik}^{(a)} = 1$ and $X_{j\ell}^{(b)} = 1$. After this transformation, the integer program consists only of binary variables, and the objective and constraints are all linear.

## A.4. Derivations of Section 3.1

For any assignment mechanism where $N_0, N_1 > 0$, the difference-in-means estimator can be written as:

$$\hat{\tau} = \sum_i Z_i \underbrace{\left( \frac{Y_i(1)}{N_1} + \frac{Y_i(0)}{N_0} \right) - \frac{Y_i(0)}{N_0}}_{a_i} \underbrace{\phantom{-\frac{Y_i(0)}{N_0}}}_{b_i} = \sum_i Z_i a_i + b_i \, .$$

where $N_1$ is the number of treated units and $N_0$ is the number of controlled units. Estimators of this form have expectation $\mathbf{E}(\hat{\tau}) = \sum_i \mathbf{E}[Z_i] a_i + b_i$ and we can write their variance as a sum over all assignments:

$$\mathbf{Var}_Z[\hat{\tau}] = \sum_z \mathbb{P}(Z = z) \left( \sum_{i=1}^{N} z_i a_i + b_i - \mathbf{E}(\hat{\tau}) \right)^2 = \sum_z \mathbb{P}(Z = z) \left( \sum_{i=1}^{N} a_i (z_i - \mathbf{E}[Z_i]) \right)^2$$

Substituting for $D_i = z_i - \mathbf{E}[Z_i]$, we have $\mathbf{Var}_Z[\hat{\tau}] = \mathbf{Var}_D[\hat{\tau}] = \sum_d \mathbb{P}(D = d) \left( \sum_i a_i d_i \right)^2 = \sum_d \mathbb{P}(D = d) \left( \sum_{i,j} d_i d_j a_i a_j \right) = \sum_{i,j} a_i a_j \mathbf{E}[D_i D_j]$, such that the objective function we are trying to maximize with respect to $Y_i(1)$ and $Y_i(0)$ is

$$\mathbf{Var}_Z[\hat{\tau}] = \sum_{i,j} a_i a_j \mathbf{Cov}[Z_i, Z_j] \tag{6}$$

## A.5. Proof of Theorems in Section 4.2

In optimization (4)(d) the slackness parameter $\epsilon$ was chosen to allow for a feasible solution. Other than feasibility, we note that while $\mathbb{E}_Z[F_a^{mis}(y)] = \mathbb{E}_Z[F_a^{obs}(y)]$, for a given sample the distributions $F_a^{mis}$ and $F_a^{obs}$ may not match exactly. In this section, we derive a finite sample analysis of optimization (4). Specifically, the next theorem provides a lower bound on the probability that for a given sample, the two distributions $F_a^{mis}$ and $F_a^{obs}$ are uniformly close to each other, namely within a distance of at most $\epsilon$. This also gives a rate for the probability of the optimal objective value of (4) upper bounding the quantity of interest $\mathbf{Var}_Z[\hat{\tau}]$, as the sample size increases.

For the readers' convenience we recall the statement of Theorem 4.1.

**Theorem A.1.** *Suppose that $Y_i(a) \sim F_a^*$ for $a \in \{0, 1\}$ and $i \in [n]$. Also assume that $\{(Y_i(0), Y_i(1))\}_{i \in [N]}$ and the assignment variables $\{Z_i\}_{i \in [N]}$ are independent. Let $V^*$ be the optimal objective value of the problem (4). We have*

$$\mathbb{P}\left( V^* \geq \mathbf{Var}_Z[\hat{\tau}] \right) \geq 1 - \beta, \tag{7}$$

*where*

$$\beta := 8 \exp\left( -\frac{\epsilon^2}{4} N \tilde{P} \right) + \frac{32}{N^2 \tilde{P}^2} \sum_{i,j \in [N]} \mathbf{Cov}(Z_i, Z_j),$$

$$\tilde{P} = \min(P, 1 - P), \quad P = \frac{1}{N} \sum_{i \in [N]} \mathbb{P}(Z_i = 1) \, .$$

*Proof.* We start by finding a lower bound for the probability that the following statement holds:

$$\forall a \in \{0, 1\}, \forall k, \; \left| \sum_{i=1}^{N} \frac{Z_i}{N_1} X_{ik}^{(a)} - F_a^*(y_k) \right| \leq \frac{\epsilon}{2}$$

Since $\mathbf{Z}$ is independent of $\mathbf{Y}^{(a)}$, the set of $\{Y_i^{(a)} : i \in [N], Z_i = 1\}$ is a set of random draws from $F_a^*$. We condition on $\mathbf{Z}$ and let $N_1 = \sum_{i=1}^{N} Z_i$. By using the Dvoretzky–Kiefer–Wolfowitz (DKW) inequality[1], we have that for every $\epsilon > 0$,

$$\mathbb{P}\left( \sup_{y_k} \left| \sum_{i=1}^{N} \frac{Z_i}{N_1} X_{ik}^{(a)} - F_a^*(y_k) \right| > \frac{\epsilon}{2} \middle| \mathbf{Z} \right) \leq 2e^{-N_1 \epsilon^2 / 2} \, .$$

---

[1]Note that DWK inequality provides a bound on the worst case distance of an empirically determined "distribution function" $\widehat{G}$ from its associated population "distribution function" $G$. For a discrete variable, we can write the pmf at a value $y$ as $F(y) = G(y+\delta) - G(y-\delta)$, for a small enough $\delta$ and apply DWK bound on $G(y + \delta)$ and $G - \delta$ separately to get a similar bound for $F(y)$.

By taking expectation with respect to $\mathbf{Z}$ we arrive at

$$\mathbb{P}\left(\sup_{y_k}\Big|\sum_{i=1}^{N}\frac{Z_i}{N_1}X_{ik}^{(a)}-F_a^*(y_k)\Big|>\frac{\epsilon}{2}\right)\leq 2\mathbb{E}[e^{-N_1\epsilon^2/2}].\tag{8}$$

The correlation among $Z_i$'s will affect the rate of convergence.

By using Chebyshev's inequality, we have

$$\mathbb{P}\Big(\Big|\sum_{i\in[N]}Z_i-\sum_{i\in[N]}P_i\Big|\geq N\delta\Big)\leq\frac{1}{N^2\delta^2}\sum_{i,j\in[N]}\mathbf{Cov}(Z_iZ_j),\tag{9}$$

with $P_i=\mathbb{P}(Z_i=1)$. Therefore,

$$\mathbb{P}\Big(N_1\leq\sum_{i\in[N]}P_i-N\delta\Big)\leq\frac{1}{N^2\delta^2}\sum_{i,j\in[N]}\mathbf{Cov}(Z_iZ_j).\tag{10}$$

We set $\delta=\frac{1}{2N}\sum_{i\in[N]}P_i$, we have

$$\mathbb{P}\Big(N_1\leq\frac{1}{2}\sum_{i\in[N]}P_i\Big)\leq\frac{4}{(\sum_i P_i)^2}\sum_{i,j\in[N]}\mathbf{Cov}(Z_i,Z_j).$$

Let $\mathcal{E}$ be the following probabilistic event:

$$\mathcal{E}:=\Big\{N_1>\frac{1}{2}\sum_{i\in[N]}P_i\Big\}.$$

Using the above bound bound on $\mathbb{P}(\mathcal{E}^c)$, we get

$$\mathbb{E}[e^{-N_1\epsilon^2/2}]\leq\exp\Big(-\frac{\epsilon^2}{4}\sum_{i\in[N]}P_i\Big)\mathbb{P}(\mathcal{E})+\mathbb{P}(\mathcal{E}^c)$$

$$\leq\exp\Big(-\frac{\epsilon^2}{4}\sum_{i\in[N]}P_i\Big)+\frac{4}{(\sum_i P_i)^2}\sum_{i,j\in[N]}\mathbf{Cov}(Z_i,Z_j).\tag{11}$$

Combining (11) and (8) we obtain the following result:

$$\mathbb{P}\Big(\sup_{y_k}\Big|\sum_{i=1}^{N}\frac{Z_i}{N_1}X_{ik}^{(a)}-F_a^*(y_k)\Big|>\frac{\epsilon}{2}\Big)\leq 2\exp\Big(-\frac{\epsilon^2}{4}\sum_{i\in[N]}P_i\Big)+\frac{8}{(\sum_i P_i)^2}\sum_{i,j\in[N]}\mathbf{Cov}(Z_i,Z_j)$$

$$=2\exp\Big(-\frac{\epsilon^2}{4}NP\Big)+\frac{8}{N^2P^2}\sum_{i,j\in[N]}\mathbf{Cov}(Z_i,Z_j).\tag{12}$$

By applying a similar argument (using $1-Z_i$ instead of $Z_i$) we also get

$$\mathbb{P}\Big(\sup_{y_k}\Big|\sum_{i=1}^{N}\frac{1-Z_i}{N_0}X_{ik}^{(a)}-F_a^*(y_k)\Big|>\frac{\epsilon}{2}\Big)\leq 2\exp\Big(-\frac{\epsilon^2}{4}N(1-P)\Big)+\frac{8}{N^2(1-P)^2}\sum_{i,j\in[N]}\mathbf{Cov}(Z_i,Z_j).\tag{13}$$

In addition, by triangle inequality we have

$$\Big|\sum_{i=1}^{N}\frac{Z_i}{N_1}X_{ik}^{(a)}-\frac{1-Z_i}{N_0}X_{ik}^{(a)}\Big|\leq\Big|\sum_{i=1}^{N}\frac{Z_i}{N_1}X_{ik}^{(a)}-F_a^*(y_k)\Big|+\Big|\sum_{i=1}^{N}\frac{1-Z_i}{N_0}X_{ik}^{(a)}-F_a^*(y_k)\Big|.$$

Therefore, by union bounding over the two events in (12) and (13) and over $a\in\{0,1\}$ we get that

$$\forall k,a,\quad\Big|\sum_{i=1}^{N}X_{ik}^{(a)}\Big(\frac{Z_i}{N_1}-\frac{1-Z_i}{N_0}\Big)\Big|\leq\epsilon,$$

with probability at least $1 - \beta$, where

$$\beta := 8 \exp\left(-\frac{\epsilon^2}{4} N\tilde{P}\right) + \frac{32}{N^2 \tilde{P}^2} \sum_{i,j \in [N]} \mathbf{Cov}(Z_i, Z_j),$$

with $\tilde{P} := \min(P, 1 - P)$.

Hence, $\mathbf{X}$ is a feasible solution to (8) with the same probability. In addition, recalling (3) we have

$$V^* = (\mathbf{X}^*)^T \mathbf{Q} \mathbf{X}^* \geq \mathbf{X}^T \mathbf{Q} \mathbf{X} = \mathbf{Var}_Z[\hat{\tau}],$$

which completes the proof. $\square$

*Remark* A.2. Under the setting of Theorem A.1, if the assignment variables $Z_i$ are independent, then we can sharpen the probability bound using Hoeffding inequality in (9). In this case, the failure probability $\beta$ will be given by

$$\beta = 8 \exp\left(-\frac{\epsilon^2}{4} N\tilde{P}\right) + 8 \exp\left(-\frac{1}{2} N\tilde{P}^2\right).$$

**Confounded independent treatment assignments.** We next prove a similar result for the case that the treatment variable can be correlated with the outcomes through a confounding factor.

For the readers' convenience we recall the statement of Theorem 4.3.

**Theorem A.3.** *Suppose that $(Y_i(0), Y_i(1), Z_i)$ are i.i.d across $i \in [n]$ and $Y_i(a) \sim F_a^*$ for $a \in \{0, 1\}$. In addition, suppose that*

$$\left| \frac{\mathbb{P}(Z = a | Y(0), Y(1))}{\mathbb{P}(Z = a)} - 1 \right| \leq \delta,$$

*for some $\delta > 0$. Note that $\delta$ controls the effect of confoundedness, with $\delta = 0$ corresponding to unconfoundedness. Let $V^*$ be the optimal objective value of the optimization problem (4), and suppose that the slackness parameter $\epsilon$, in constraint (d) satisfies $\epsilon \geq \delta$. We have*

$$\mathbb{P}\left(V^* \geq \mathbf{Var}_Z[\hat{\tau}]\right) \geq 1 - \beta, \tag{14}$$

*where*

$$\beta := 1 - 8 e^{-2N\epsilon_0^2} - 8 e^{-N\tilde{P}\epsilon_1^2/3},$$

$$\tilde{P} = \min(P, 1 - P), \quad \epsilon_0 = \frac{(\epsilon - \delta)\tilde{P}(1 + \delta)}{2 + \delta + \epsilon}, \quad \epsilon_1 = \frac{\epsilon - \delta}{2 + \delta + \epsilon}.$$

*Proof.* Recall that $X_{ik}^{(a)} := \mathbb{I}(Y_i(a) = y_k)$. We first establish the following lemma.

**Lemma A.4.** *Under the assumption of Theorem A.3 we have*

$$\left| \frac{\mathbb{P}(Z_i = a | X_{ik}^{(a)})}{\mathbb{P}(Z_i = a)} - 1 \right| \leq \delta.$$

We define the probabilistic event $\mathcal{E}_a$ as follows, for a fixed arbitrary $\epsilon_0 > 0$:

$$\mathcal{E}_a := \left\{ \sup_{y_k} \left| \sum_{i=1}^N \frac{Z_i}{N} X_{ik}^{(a)} - \mathbb{P}(Z = 1 | X_k^{(a)}) \, F_a^*(y_k) \right| \leq \epsilon_0 \right\}.$$

Note that by assumption $Z_i | X_{ik}^{(a)}$ are i.i.d and we denote their distribution by dropping the index $i$ as $Z | X_k^{(a)}$.

By an application of Dvoretzky–Kiefer–Wolfowitz (DKW) inequality, similar to the proof of Theorem A.1, we have $\mathbb{P}(\mathcal{E}_a) \leq 2 e^{-2N\epsilon_0^2}$.

We also define the probabilistic event $\widetilde{\mathcal{E}}$ as follows:

$$\widetilde{\mathcal{E}} := \left\{ \left| \frac{1}{N} \sum_{i=1}^{N} Z_i - \mathbb{P}(Z=1) \right| \leq \epsilon_1 \, \mathbb{P}(Z=1) \right\} . \tag{15}$$

Recall the notation $N_1 := \sum_{i=1}^{N} Z_i$ and let $P = \mathbb{P}(Z=1)$. Then, by an application of the multiplicative Chernoff bound we have $\mathbb{P}(\widetilde{\mathcal{E}}) \leq 2e^{-NP\epsilon_1^2/3}$.

Therefore, on the event $\mathcal{E} \cap \widetilde{\mathcal{E}}$, we have

$$\sum_{i=1}^{N} \frac{Z_i}{N_1} X_{ik}^{(a)} = \frac{\sum_{i=1}^{N} Z_i X_{ik}^{(a)}}{\sum_{i=1}^{N} Z_i}$$

$$\leq \frac{\left| \sum_{i=1}^{N} \frac{Z_i}{N} X_{ik}^{(a)} - \mathbb{P}(Z=1|X_k^{(a)})F_a^*(y_k) \right| + \mathbb{P}(Z=1|X_k^{(a)})F_a^*(y_k)}{\mathbb{P}(Z=1) - \left| \mathbb{P}(Z=1) - \frac{1}{N}\sum_{i=1}^{N} Z_i \right|}$$

$$\leq \frac{\epsilon_0 + \mathbb{P}(Z=1|X_k^{(a)})F_a^*(y_k)}{P(1-\epsilon_1)}$$

$$\leq \frac{\epsilon_0}{P(1-\epsilon_1)} + \frac{1+\delta}{1-\epsilon_1} F_a^*(y_k) ,$$

where in the last step we used the result of Lemma A.4, by which $\mathbb{P}(Z=1|X_k^{(a)})/P \leq 1+\delta$.

Likewise, on the event $\mathcal{E} \cap \widetilde{\mathcal{E}}$, we have

$$\sum_{i=1}^{N} \frac{Z_i}{N_1} X_{ik}^{(a)} \geq \frac{\mathbb{P}(Z=1|X_k^{(a)})F_a^*(y_k) - \epsilon_0}{P(1+\epsilon_1)}$$

$$\geq \frac{1-\delta}{1+\epsilon_1} F_a^*(y_k) - \frac{\epsilon_0}{P(1+\epsilon_1)} ,$$

Therefore, on the event $\mathcal{E} \cap \widetilde{\mathcal{E}}$, we have

$$\sup_{y_k} \left| \sum_{i=1}^{N} \frac{Z_i}{N_1} X_{ik}^{(a)} - F_a^*(y_k) \right|$$

$$\leq \max \left\{ \frac{\epsilon_0}{P(1-\epsilon_1)} + \left( \frac{1+\delta}{1-\epsilon_1} - 1 \right) F_a^*(y_k), \frac{\epsilon_0}{P(1+\epsilon_1)} + \left( 1 - \frac{1-\delta}{1+\epsilon_1} \right) F_a^*(y_k) \right\}$$

$$\leq \frac{\epsilon_0}{P(1-\epsilon_1)} + \frac{\epsilon_1+\delta}{1-\epsilon_1} ,$$

since $F_a^*(y_k) \leq 1$. In addition, by union bounding,

$$\mathbb{P}(\mathcal{E} \cap \widetilde{\mathcal{E}}) \geq 1 - \mathbb{P}(\mathcal{E}^c) - \mathbb{P}(\widetilde{\mathcal{E}}^c) \geq 1 - 2e^{-2N\epsilon_0^2} - 2e^{-NP\epsilon_1^2/3} .$$

By following a similar argument (replacing the role of $Z_i$ by $1-Z_i$), we obtain that

$$\sup_{y_k} \left| \sum_{i=1}^{N} \frac{1-Z_i}{N_0} X_{ik}^{(a)} - F_a^*(y_k) \right| \leq \frac{\epsilon_0}{(1-P)(1-\epsilon_1)} + \frac{\epsilon_1+\delta}{1-\epsilon_1} ,$$

with probability at least $1 - 2e^{-2N\epsilon_0^2} - 2e^{-N(1-P)\epsilon_1^2/3}$.

Therefore, by triangle inequality and union bounding over $a \in \{0,1\}$ we get that

$$\forall k, a, \quad \left| \sum_{i=1}^{N} X_{ik}^{(a)} \left( \frac{Z_i}{N_1} - \frac{1-Z_i}{N_0} \right) \right| \leq 2 \left[ \frac{\epsilon_0}{\tilde{P}(1-\epsilon_1)} + \frac{\epsilon_1+\delta}{1-\epsilon_1} \right] , \tag{16}$$

with probability at least $1 - \beta$, where

$$\beta := 1 - 8e^{-2N\epsilon_0^2} - 8e^{-N\tilde{P}\epsilon_1^2/3}$$

and $\tilde{P} = \min(P, 1 - P)$.

We next set $\epsilon_0$, $\epsilon_1$ such that the right-hand side of (16) becomes $\epsilon$. Namely, we set

$$\epsilon_0 = \frac{(\epsilon - \delta)\tilde{P}(1 + \delta)}{2 + \delta + \epsilon}, \quad \epsilon_1 = \frac{\epsilon - \delta}{2 + \delta + \epsilon}.$$

This way, $\mathbf{X}$ is a feasible solution to (8) with probability at least $1 - \beta$, and so

$$V^* = (\mathbf{X}^*)^T \mathbf{Q} \mathbf{X}^* \geq \mathbf{X}^T \mathbf{Q} \mathbf{X} = \mathbf{Var}_Z[\hat{\tau}].$$

$\square$

*Proof.* (Proof of Lemma A.4) By definition, $\mathbb{P}(Z_i = a | X_{ik}^{(a)} = 1) = \mathbb{P}(Z_i = a | Y_i(a) = y_k)$. Therefore, from the assumption we have

$$\left| \frac{\mathbb{P}(Z_i = a | X_{ik}^{(a)} = 1)}{\mathbb{P}(Z_i = a)} - 1 \right| \leq \delta.$$

For the other case we write

$$\mathbb{P}(Z_i = 1 | X_{ik}^{(a)} = 0) = \frac{\mathbb{P}(Z_i = 1, X_{ik}^{(a)} = 0)}{\mathbb{P}(X_{ik}^{(a)} = 0)} = \frac{\sum_{\ell \neq k} \mathbb{P}(Z_i = 1 | Y_i(a) = y_\ell)\mathbb{P}(Y_i(a) = y_\ell)}{\mathbb{P}(X_{ik}^{(a)} = 0)}$$

Expanding the denominator as $\mathbb{P}(X_{ik}^{(a)} = 0) = \sum_{\ell \neq k} \mathbb{P}(Y_i(a) = y_\ell)$, we get

$$\frac{\mathbb{P}(Z_i = 1 | X_{ik}^{(a)} = 0)}{\mathbb{P}(Z_i = a)} = \frac{\sum_{\ell \neq k} \mathbb{P}(Z_i = 1 | Y_i(a) = y_\ell)\mathbb{P}(Y_i(a) = y_\ell)}{\sum_{\ell \neq k} \mathbb{P}(Z_i = a)\mathbb{P}(Y_i(a) = y_\ell)}.$$

Define the shorthand $a_\ell := \mathbb{P}(Z_i = 1 | Y_i(a) = y_\ell)\mathbb{P}(Y_i(a) = y_\ell)$ and $b_\ell := \mathbb{P}(Z_i = a)\mathbb{P}(Y_i(a) = y_\ell)$. Then by our assumption $|\frac{a_\ell}{b_\ell} - 1| \leq \delta$, or equivalently, $(1 - \delta)b_\ell \leq a_\ell \leq (1 + \delta)b_\ell$. This implies that

$$1 - \delta \leq \frac{\mathbb{P}(Z_i = 1 | X_{ik}^{(a)} = 0)}{\mathbb{P}(Z_i = a)} = \frac{\sum_{\ell \neq k} a_\ell}{\sum_{\ell \neq k} b_\ell} \leq 1 + \delta.$$

This concludes the proof of lemma. $\square$

### A.6. Derivations of Section 3.3

We first prove that the variance of estimators of the form $\hat{\tau} = \sum_i b_i + \sum_j Z_i Z_j a_{ij}$ can be written as a sum of $N^4$ products of $a_{ij}$ and $a_{kl}$ terms:

$$\mathbf{Var}_Z[\hat{\tau}] = \sum_z \mathbb{P}(Z = z) \left( \sum_i b_i + \sum_j Z_i Z_j a_{ij} - \sum_i b_i - \sum_j \mathbb{E}[Z_i Z_j] a_{ij} \right)^2$$

$$= \sum_z \mathbb{P}(Z = z) \left( \sum_{i,j} (Z_i Z_j - \mathbb{E}[Z_i Z_j]) a_{ij} \right)^2$$

$$= \sum_{i,j,k,l} a_{ij} a_{kl} \mathbf{Cov}[Z_i Z_j, Z_k Z_l]$$

We now prove that the regression estimator on $Z$ and a covariate $X$ (later rewritten to $W$ to avoid confusion with the indicator variables) can be parameterized in such a way.

Suppose that $f$ is a linear regression with covariate $X$. The exact model is $Y_i = \beta_0 + \beta_1 D_i + \beta_2 X_i$, where $D_i = Z_i - \frac{N_1}{N}$. $\beta_1$ gives the average treatment effect. Without loss of generality, assume $\frac{1}{N} \sum_i X_i = 0$ (any nonzero mean can be absorbed into the intercept). We can analytically compute the hat matrix, which is simplified by the fact that $\sum_i D_i = 0$ and $\sum_i X_i = 0$.

$$X^T X = \begin{bmatrix} N & 0 & 0 \\ 0 & \frac{N_1 N_0}{N} & \sum_i X_i D_i \\ 0 & \sum_i X_i D_i & \sum_i (X_i^2) \end{bmatrix}$$

We compute the inverse of this matrix using Gaussian elimination. Because we only need the second row of the inverse, it suffices to only partially solve the Gaussian elimination. We only compute the first two rows of the inverse:

$$(X^T X)^{-1} = \begin{bmatrix} \frac{1}{N} & 0 & 0 \\ 0 & \frac{N}{N_1 N_0} & -\frac{N}{N_1 N_0} \cdot \frac{\sum_i X_i D_i}{\sum_i (X_i^2)} \\ 0 & ? & ? \end{bmatrix}$$

The average treatment effect estimate $\hat{\beta}_1$ is given by

$$\begin{aligned}
\hat{\tau} = [(X^T X)^{-1} X^T Y^{obs}]_2 &= \begin{bmatrix} 0 & \frac{N}{N_1 N_0} & -\frac{N}{N_1 N_0} \cdot \frac{\sum_i X_i D_i}{\sum_i (X_i^2)} \end{bmatrix} \begin{bmatrix} \sum_i Y_i^{obs} \\ \sum_i Y_i^{obs} D_i \\ \sum_i Y_i^{obs} X_i \end{bmatrix} \\
&= \frac{N}{N_1 N_0} \cdot \left( \sum_i Y_i^{obs} D_i \right) - \frac{N}{N_1 N_0} \cdot \left( \frac{\sum_i X_i D_i}{\sum_i (X_i^2)} \cdot \sum_i Y_i^{obs} X_i \right) \\
&= \frac{N}{N_1 N_0} \sum_i D_i \left( Y_i^{obs} - X_i \frac{\sum_j X_j Y_j^{obs}}{\sum_j X_j^2} \right)
\end{aligned}$$

Next, we perform two algebraic manipulations to write our estimator in terms of the potential outcomes (instead of $Y_i^{obs}$).

$$\begin{aligned}
Y_i^{obs} &= Z_i Y_i(1) + (1 - Z_i) Y_i(0) =: Z_i \hat{Y}_i + Y_i(0) \\
D_i Y_i^{obs} &= \left( Z_i - \frac{N_1}{N} \right) (Z_i Y_i(1) + (1 - Z_i) Y_i(0)) \\
&= Z_i Y_i(1) - \frac{N_1}{N} Z_i Y_i(1) - \frac{N_1}{N} Y_i(0) + Z_i Y_i(0) \frac{N_1}{N} \\
&= Z_i \left( Y_i(1) \frac{N_0}{N} + Y_i(0) \frac{N_1}{N} \right) - \frac{N_1}{N} Y_i(0) \\
&= Z_i \frac{N_1 N_0}{N} \left( \frac{Y_i(1)}{N_1} + \frac{Y_i(0)}{N_0} \right) - \frac{N_1}{N} Y_i(0) =: Z_i \frac{N_1 N_0}{N} \check{Y}_i - \frac{N_1}{N} Y_i(0)
\end{aligned}$$

where we introduce the notation $\hat{Y}_i := Y_i(1) - Y_i(0)$ and $\check{Y}_i := \frac{Y_i(1)}{N_1} + \frac{Y_i(0)}{N_0}$.

$$\hat{\tau} = \frac{N}{N_1 N_0} \sum_i \left( D_i Y_i^{obs} - \frac{1}{\mathbf{X}^T \mathbf{X}} \sum_j D_i X_i X_j Y_j^{obs} \right)$$

$$= \frac{N}{N_1 N_0} \sum_i \left( Z_i \frac{N_1 N_0}{N} \check{Y}_i - \frac{N_1}{N} Y_i(0) - \frac{1}{\mathbf{X}^T \mathbf{X}} \sum_j D_i X_i X_j (Z_j \hat{Y}_j + Y_j(0)) \right)$$

$$= \frac{N}{N_1 N_0} \sum_i \left( -\frac{N_1}{N} Y_i(0) \right) + \frac{N}{N_1 N_0} \sum_i \left( Z_i \frac{N_1 N_0}{N} \check{Y}_i - \frac{1}{\mathbf{X}^T \mathbf{X}} \sum_j (Z_i - \frac{N_1}{N}) X_i X_j (Z_j \hat{Y}_j + Y_j(0)) \right)$$

$$= -\frac{1}{N_0} \sum_i Y_i(0)$$

$$+ \frac{N}{N_1 N_0} \frac{1}{\mathbf{X}^T \mathbf{X}} \sum_i \sum_j \left( Z_i \mathbf{X}^T \mathbf{X} \frac{N_1 N_0}{N^2} \check{Y}_i - X_i X_j \left( Z_i Z_j \hat{Y}_j + Z_i Y_j(0) - \frac{N_1}{N} Z_j \hat{Y}_j - \frac{N_1}{N} Y_j(0) \right) \right)$$

$$= -\frac{1}{N_0} \sum_i Y_i(0) - \frac{1}{N_0} \frac{1}{\mathbf{X}^T \mathbf{X}} \sum_i \sum_j X_i X_j Y_j(0)$$

$$+ \frac{N}{N_1 N_0} \frac{1}{\mathbf{X}^T \mathbf{X}} \sum_i \sum_j \left( Z_i \mathbf{X}^T \mathbf{X} \frac{N_1 N_0}{N^2} \check{Y}_i - X_i X_j \left( Z_i Z_j \hat{Y}_j + Z_i Y_j(0) - \frac{N_1}{N} Z_j \hat{Y}_j \right) \right)$$

It follows that

$$\hat{\tau} = \sum_i \underbrace{-\frac{1}{N_0} \left( Y_i(0) X_i \frac{\mathbf{1}^T \mathbf{X}}{\mathbf{X}^T \mathbf{X}} \right)}_{b_i} +$$

$$+ \frac{N}{N_1 N_0} \frac{1}{\mathbf{X}^T \mathbf{X}} \sum_i \sum_j \left( Z_i \mathbf{X}^T \mathbf{X} \frac{N_1 N_0}{N^2} \check{Y}_i - X_i X_j \left( Z_i Z_j \hat{Y}_j + Z_i Y_j(0) - \frac{N_1}{N} Z_j \hat{Y}_j \right) \right)$$

Because the double sum contains all combinations of $i$ and $j$, we can reindex $i$ and $j$ in the term $\frac{N_1}{N} Z_j \hat{Y}_j X_i X_j$:

$$\hat{\tau} = \sum_i b_i + \frac{N}{N_1 N_0} \frac{1}{\mathbf{X}^T \mathbf{X}} \sum_i \sum_j \left( Z_i \mathbf{X}^T \mathbf{X} \frac{N_1 N_0}{N^2} \check{Y}_i - X_i X_j \left( Z_i Z_j \hat{Y}_j + Z_i Y_j(0) - \frac{N_1}{N} Z_i \hat{Y}_i \right) \right)$$

$$= \sum_i b_i + \frac{N}{N_1 N_0} \frac{1}{\mathbf{X}^T \mathbf{X}} \sum_i \sum_j \left( Z_i \left( \mathbf{X}^T \mathbf{X} \frac{N_1 N_0}{N^2} \check{Y}_i - X_i X_j Y_j(0) + \frac{N_1}{N} \hat{Y}_i \right) - Z_i Z_j X_i X_j \hat{Y}_j \right)$$

We see we can write $\hat{\tau} = \sum_i b_i + \sum_i \sum_j a_{ij} Z_i Z_j$, where

$$b_i = -\frac{1}{N_0} \left( Y_i(0) + X_i \frac{\mathbf{1}^T \mathbf{X}}{\mathbf{X}^T \mathbf{X}} \right)$$

$$a_{ij} = \begin{cases} \frac{1}{N} \check{Y}_i - \frac{N}{N_1 N_0} \frac{1}{\mathbf{X}^T \mathbf{X}} X_i X_j Y_j(0) + \frac{1}{N_0} \frac{1}{\mathbf{X}^T \mathbf{X}} \hat{Y}_i & \text{if } i = j \\ -\frac{N}{N_1 N_0} \frac{1}{\mathbf{X}^T \mathbf{X}} X_i X_j \hat{Y}_j & \text{otherwise} \end{cases}$$

Substituting in our definitions of $\check{Y}_i$ and $\hat{Y}_i$, we have

$$b_i = -\frac{1}{N_0} \left( Y_i(0) + X_i \frac{\mathbf{1}^T \mathbf{X}}{\mathbf{X}^T \mathbf{X}} \right)$$

$$a_{ij} = \begin{cases} \frac{1}{N_1 N} Y_i(1) + \frac{1}{N_0 N} Y_i(0) - \frac{N}{N_1 N_0} \frac{1}{\mathbf{X}^T \mathbf{X}} X_i X_j Y_j(0) + \frac{1}{N_0} \frac{1}{\mathbf{X}^T \mathbf{X}} (Y_i(1) - Y_i(0)) & \text{if } i = j \\ -\frac{N}{N_1 N_0} \frac{1}{\mathbf{X}^T \mathbf{X}} X_i X_j (Y_i(1) - Y_i(0)) & \text{otherwise} \end{cases}$$

### A.7. Conservative Variance Bounds for Section 5

Here we make explicit the analytical, conservative variance estimators used as baselines for the complete randomization and matched pairs designs in Section 5. Recall that $N_0$, $N_1$ are the number of units in the control and treatment sets, respectively,

that $N = N_0 + N_1$ is the total number of units, and that $Z_i \in \{0, 1\}$ is the treatment indicator for unit $i$. For $z \in \{0, 1\}$, we define the sample variance estimator

$$s_z^2 = \tfrac{1}{N_z - 1} \sum_{i: Z_i = z} \left( Y_i(z) - \tfrac{1}{N_z} \sum_{i: Z_i = z} Y_i(z) \right)^2.$$

An estimator for the sampling variance of diff-in-means under complete randomization is as follows; see Theorem 6.3 in (Imbens & Rubin, 2015).

$$\frac{s_0^2}{N_0} + \frac{s_1^2}{N_1}.$$

Comparing to (1), we see that this estimator is a conservative upper bound for the true sampling variance as $S_\tau^2 \geq 0$.

For the matched pairs design, let $\hat{\tau}(j)$ be the difference between the outcomes for the treatment and control units in pair $j$, for $j \in [N/2]$. A conservative estimator for the sampling variance of diff-in-means under the matched pairs design is as follows; see Theorem 10.1 in (Imbens & Rubin, 2015).

$$\frac{4}{N(N-2)} \sum_{j=1}^{N/2} \left( \hat{\tau}(j) - \frac{1}{N/2} \sum_{j=1}^{N/2} \hat{\tau}(j) \right).$$

### A.8. Variance of the Difference-in-means Estimator under a Completely Randomized Assignment

The variance of the difference-in-means estimator is given by

$$\mathbf{Var}_Z[\hat{\tau}_{DIM}] = \frac{S_1^2}{N_1} + \frac{S_0^2}{N_0} - \frac{S_\tau^2}{N}, \tag{17}$$

where $S_z^2 = \frac{1}{N-1} \sum_{i=1}^N \left( Y_i(z) - \frac{1}{N} \sum_{i=1}^N Y_i(z) \right)^2$ is the sample variances of the treated ($z = 1$) and control ($z = 0$) units, while $S_\tau^2 := \frac{1}{N-1} \sum_{i=1}^N \left( Y_i(1) - Y_i(0) - \frac{1}{N} \sum_{i=1}^N (Y_i(1) - Y_i(0)) \right)^2$ is the sample variance of the unit-level treatment effects $Y_i(1) - Y_i(0)$.

### A.9. Constructing confidence intervals

We propose two different ways of constructing confidence intervals from the optimal value $V^*$ of the quadratic program above. The first approach is to use a normal asymptotic approximation. In this case, the $(1 - \alpha)$-confidence interval is computed as $\hat{\tau} \pm (\sqrt{V^*/N}) \cdot z_{1-\alpha/2}$ with $z_{1-\alpha/2}$ being the $(1 - \alpha/2)$-quantile of the standard normal distribution. This approach is justified in some settings. In particular, Abadie, Athey, Imbens, and Wooldridge (2020) study the asymptotic validity of this approach for linear regression. In other settings, it can be viewed as an approximation, the properties of which can be investigated by running simulations or A/A-tests. Another approach is to employ a bootstrap procedure similar to the one proposed in Imbens & Menzel (2021), in which unobserved potential outcomes are imputed via the least favorable copula and bootstrap samples are drawn via rerandomization of units to treatment. Confidence intervals are constructed based on the quantiles of the bootstrap distribution of $\hat{\tau}$, as illustrated in Algorithm 1. Many variants of the bootstrap can also be applied to a causal bootstrap; for example Imbens and Menzel (2021) bootstrap the $t$-statistic which is asymptotically pivotal and is known to lead to an asymptotic refinement of the confidence interval in standard settings (Hall, 2013).

In Algorithm 1, $\mathrm{supp}(F_a)$ is the support of the marginal distributions of outcomes $F_a$. If $\mathrm{supp}(F_a)$ is not known (e.g. continuous outcomes), we can use the support of the observed marginal distributions $\mathrm{supp}(F_a^{obs})$ in Algorithm 1. We revisit the validity of this assumption in 4.2, and the specific parameterization of the solution $\mathbf{X}$ in section 2.4.

### A.10. Understanding the undercoverage of bootstrap-based methods

Copula-based methods are liable to suffer from undercoverage in small samples. This is true of the isotone copula method proposed by Aronow et al. (2014) and used by Imbens & Menzel (2021), as well as our own method. These concerns diminish in larger samples. Consider the following scenario: The true effect is 0. Regardless of the realized randomization,

---

**Algorithm 1** Our Causal Bootstrap Procedure

---

**Require:** for $a \in \{0, 1\}$ : $\mathbf{y(a)}$, vector representation of $\mathrm{supp}(F_a)$; $\mathbf{X}^{(a)}$, solution to Eq. 2; design $\mathcal{P}(\mathbf{Z})$.
**Initialize:** $S \leftarrow \emptyset$
**for** $a \in \{0, 1\}$ **do**
    $\mathbf{Y}(a) \leftarrow (\mathbf{y^{(a)}})^{\mathbf{T}}\mathbf{X^{(a)}}$;
**end for**
**for** some large number of repetitions **do**
    Sample $\mathbf{Z} \sim \mathcal{P}(\mathbf{Z})$ and $S \leftarrow S \cup \{\hat{\tau}(\mathbf{Z}, \mathbf{Y}(a))\}$
**end for**
**Return** $[q_{\alpha/2}, q_{1-\alpha/2}]$ ($q_x$ is the $x^{th}$ quantile of $S$).

---

| Unit | Pair | $Y_0$ | $Y_1$ |
|------|------|-------|-------|
| A | 1 | 11 | 11 |
| B | 1 | 10 | 10 |
| C | 2 | 0 | 0 |
| D | 2 | 0 | 0 |

any copula that matches the marginals will lead to a non-zero ATE deterministically. For example, in each of the observed potential outcome tables below,

| Unit | Pair | $Y_0$ | $Y_1$ |
|------|------|-------|-------|
| A | 1 | X | 11 |
| B | 1 | 10 | X |

| Unit | Pair | $Y_0$ | $Y_1$ |
|------|------|-------|-------|
| A | 1 | 11 | X |
| B | 1 | X | 10 |

$\dots$

the copula will always be imputed as either:

| Unit | Pair | $Y_0$ | $Y_1$ |
|------|------|-------|-------|
| A | 1 | 10 | 11 |
| B | 1 | 10 | 11 |

| Unit | Pair | $Y_0$ | $Y_1$ |
|------|------|-------|-------|
| A | 1 | 11 | 10 |
| B | 1 | 11 | 10 |

In either case, the samples of the estimated ATE will be $0.5$ or $-0.5$ deterministically, leading to a coverage of zero. These issues disappear in larger samples. We refer the reader to our asymptotic validity results for theoretical guarantees in this setting.

