# OpenReview forum: "Integer Programming for Generalized Causal Bootstrap Designs"
_ICML.cc/2025/Conference — ICML 2025 poster_

### Official Review · Reviewer_5o6Q · 2025-03-12

**Overall Recommendation:** 4

**Summary:**

The authors proposed to numerically estimate the joint distribution with the highest variance thus leading to the ATE estimate with the lowest variance for an RCT.

Their optimization problem is modeled by all the posible choices of assignment rules but instead of optimizing over all possible assignments they optimize over all possible potential outcomes for the sample observed. This modification leads to the constraints where there is only one posible potential outcome per observation and its consistency with the ascertainment variable Z. Additionally for a random assignment the marginals of the observed vs missed outcomes must match which add an extra constraint which the authors relax a bit in order to get a more stable program. However this last constraint is relaxed to allow for general assignment probability.

The authors then proceed to prove asymptotic validity of their methods produce a true bound bye ensuring that the true distribution is always a feasible solution. They derive a high probability bound on the variance that depends on the slack factor and the sample size. the result is valid even in the case of individual confounded assignment.

Finally simulators rustles are presented.

Weakness
- Is a computationally expensive approach.
- The guarantees are for the variance, but there is no guarantees on the ATE.

Strengths
- The authors produce finite sample guarantees for their method.
- The authors extend their method to a wide variety of estimators.

Overall a very strong work

**Claims And Evidence:**

The theory is sound and the experiments back up the results obtained.

**Essential References Not Discussed:**

I am not well versed in the literature closely related to this paper.

**Experimental Designs Or Analyses:**

Refer to the summary

**Methods And Evaluation Criteria:**

Refer to the summary

**Other Comments Or Suggestions:**

none

**Other Strengths And Weaknesses:**

refer to the Summary.

**Questions For Authors:**

- Besides the variance are there any asymptotic guarantees for the ATE? How the computed ATE from the sample obtained as the minimizer relates with the true ATE?

**Relation To Broader Scientific Literature:**

I am not well versed in the literature closely related to this paper.

**Theoretical Claims:**

The math is sound

---

> ### Author Rebuttal · Authors · 2025-04-01
>
> We thank the reviewer for their question about theoretical guarantees on the ATE. Our results all bound the variance of the ATE, so we believe the reviewer is asking about the bias of the ATE. Our results hold for a general class of quadratic-in-treatment estimators, which each have their own bias characteristics. This class includes estimators that are unbiased under SUTVA, such as Horvitz-Thompson. The class also includes a wide range of biased estimators, such as "any unbiased estimator plus a constant offset." Because of the generality of our estimator class, we do not aim to provide general characterizations of the bias of the ATE.

---

### Official Review · Reviewer_bxyv · 2025-03-21

**Overall Recommendation:** 3

**Summary:**

This paper proposed a new integer program to jointly address two sources of uncertainty in causal inference, the design uncertainty due to the treatment assignment mechanism, and sampling uncertainty. Traditional methods tend to address one of the two uncertainties, but do not handle them at the same time. Motivated by this gap, the paper proposed an integer program formulation which computes numerically the worst-case copula used as an input to the causal bootstrap method. Further, the paper proved the asymptotic validity of this method for unconfounded, conditionally unconfounded, and individualistic with bounded confoundedness assignments. Numerical experiments support the effectiveness of the proposed methodology.

### update after rebuttal:
I have read through the authors' response and updated my scores accordingly.

**Claims And Evidence:**

Overall the paper stated its claims and evidence clearly.
- Theoretically, the proposed integer program aims to identify the joint potential outcome distribution that maximizes the variance of the chosen estimator, while being consistent with the randomization design and the observed marginal distributions of potential outcomes. The proposed optimization's objective and constraints were described in detail in Section 2 for the basic difference-in-means estimator.
- On the statistical validity front, the paper provided a rate for the probability that the proposed method upper-bounds the true variance in large sample limit, by bounding the probability that the observed and missing marginal distributions are uniformly within epsilon distance.
- In the special case of conditionally unconfounded assignments, the paper shows that one can further imposes equality of the conditional marginal distributions to fully utilize the covariate information.
The main claims were supported by convincing evidence.

**Essential References Not Discussed:**

Essential references were discussed in this work.

**Experimental Designs Or Analyses:**

I did not find particular issues with the experimental design.

**Methods And Evaluation Criteria:**

The proposed method matters the most in settings with small fixed samples and heterogeneous treatment effects, such as in geographical experiments. Thus the paper used the GDP data report by IMF as a real-world geographical dataset. The proposed procedure were compared with three baselines including standard bootstrap and causal bootstrap proposed in prior works. The dataset and evaluation criteria are reasonable.

**Other Comments Or Suggestions:**

typos: "and and" in abstract

**Other Strengths And Weaknesses:**

The experiment section is relatively weak with one single dataset and limited comparisons. It will strengthen the work if a larger dataset and more baselines can be used for stress testing and comparisons.

**Questions For Authors:**

See comments above.

**Relation To Broader Scientific Literature:**

The proposed integer programming is novel and interesting as in the broader literature, which brings a gap between the design uncertainty and sampling uncertainty.

**Theoretical Claims:**

Checked through section 2's theoretical Claims, including proofs in appendix A.2 - for which I did not find issue with their correctness.

---

> ### Author Rebuttal · Authors · 2025-04-01
>
> We thank the reviewer for their careful consideration of our work and overall positive review. The main concern seems to be with the number of datasets and baselines. We were unfortunately limited by space considerations, and relegated additional experimental results to Appendix A.1. We are excited about evaluating this method in a wide variety of contexts against many possible baselines, though we believe a truly comprehensive evaluation is best left to future work given the current space constraints.

---

### Official Review · Reviewer_XXTL · 2025-03-21

**Overall Recommendation:** 3

**Summary:**

This paper proposes a novel method for quantifying **design uncertainty** in causal inference settings, particularly when experiments involve small samples, heterogeneous treatment effects, or non-standard assignment mechanisms. The standard bootstrap only captures sampling uncertainty, while existing causal bootstrap methods are limited to completely randomized designs and simple estimators like difference-in-means. The authors generalize causal bootstrap by formulating the worst-case copula problem as an **integer programming** (IP) task, applicable to a broad class of estimators (linear- and quadratic-in-treatment) and assignment mechanisms (including unconfounded, conditionally unconfounded, and bounded confoundedness cases). They prove asymptotic validity of the approach, and showcase improved variance estimates and tighter confidence intervals on simulated geographical experiments using IMF data. The approach enables exact variance maximization without closed-form copulas, providing flexibility and rigor for real-world experimental designs.

**Claims And Evidence:**

The main claims are:
- Existing causal bootstrap methods are limited in scope; integer programming enables generalization to arbitrary estimators and assignment designs.
- The proposed method provides valid, tighter confidence intervals across a range of realistic designs and estimators.
- The method remains asymptotically valid even under bounded confoundedness and conditional unconfoundedness.

These claims are strongly supported:
- The integer program is precisely formulated with all necessary constraints and relaxation mechanisms.
- The theoretical results (Lemmas and Theorems 4.1–4.3) provide probabilistic bounds and convergence rates.
- Simulations across additive and multiplicative effects, covariate scenarios, and design types (CR, matched pairs) validate the empirical advantages.

**Essential References Not Discussed:**

Most relevant references are covered, including:
- Neyman, Aronow, Imbens, Robins, Harshaw, Ji et al., and causal bootstrap literature

One possible addition: more discussion of connections to **causal bounds via OT or copula methods**, such as:
- Ji et al. (2023, AISTATS) used dual OT but not explicit worst-case coupling
- Literature on robust causal bounds under partial identification (e.g., Manski-style or Fan & Park (2010))

However, this is a minor omission.

**Experimental Designs Or Analyses:**

The experiments are well designed:
- Real GDP data is used to simulate geographical treatment effects
- Both CR and matched-pairs designs are compared
- Outcome models include both additive and multiplicative effects
- Covariate information is incorporated through doubly robust estimation and growth modeling

CI width, power, and coverage are all reported. Tables are clear and informative. Solver runtimes and scalability are also discussed.

**Methods And Evaluation Criteria:**

The core methodology—variance maximization via an integer program under potential outcome and assignment constraints—is well-motivated and mathematically sound. The paper systematically builds from the classic Neyman decomposition to modern extensions, supporting:
- Arbitrary potential outcome distributions via discretization
- Arbitrary linear- and quadratic-in-treatment estimators
- Known and probabilistic assignment mechanisms (CR, Bernoulli, matched-pairs)

Evaluation is appropriate:
- Experiments simulate realistic geographical designs
- Estimators include difference-in-means and doubly robust estimators
- Baselines include sampling bootstrap, conservative variance, and isotone copula

**Other Comments Or Suggestions:**

NA

**Other Strengths And Weaknesses:**

**Strengths:**
- Elegant use of integer programming to generalize causal inference tools
- Applicable to a wide range of estimators and designs
- Excellent theoretical backing and empirical validation
- Readable and well-structured exposition

**Weaknesses:**
- Integer programming scalability limits application to large-scale datasets (>1000 units)
- Requires discretization of outcome support, which may lead to approximation error
- Some approximations (e.g., empirical matching of marginals) may be fragile in heavy-tailed or sparse data settings

**Questions For Authors:**

1. **Scalability to Large-Scale Experiments:**
   Your method is elegant but computationally heavy. Do you foresee any way to scale to thousands or millions of units (e.g., through greedy relaxation, LP relaxations, stochastic approximations)?

2. **Robustness to Outcome Discretization:**
   Discretizing continuous outcomes is a strong assumption. How sensitive are your variance bounds and coverage to discretization granularity?

3. **Adaptive Grid Design:**
   Have you considered adaptive or non-uniform binning (e.g., quantile bins) to improve efficiency or accuracy of the integer program formulation?

4. **Extension to Clustered Designs:**
   Can your approach be extended to clustered randomizations or interference settings, where design uncertainty is entangled with spillover effects?

5. **Automated Constraint Tuning (ε):**
   Is there a principled way to select or adaptively shrink the marginal balance slackness ε, beyond theoretical feasibility guarantees?

6. **Limitations of Matched-Pair Analysis:**
   Your method identifies matched-pair isotone copula bootstrap as degenerate. Are there better imputations in that regime (e.g., hierarchical Bayesian)?

7. **Alternative Optimization Frameworks:**
   Why integer programming instead of convex relaxations or dual OT (e.g., Kantorovich dual)? Is IP strictly necessary to preserve optimality?

8. **Comparison to Copula-Based Bounds (Fan & Park, 2010):**
   How does your method relate in tightness or assumptions to classic copula-based bounds, or nonparametric bound estimators in econometrics?

9. **Multi-valued Treatments and Interaction Effects:**
   Could the integer program be extended to handle multi-valued treatments, interactions, or factorial designs beyond binary assignments?

10. **Real-World Adoption and Open-Source Tools:**
   Are there plans to release user-friendly packages or APIs for this method? What feedback have you received from practitioners in A/B testing platforms or policy experiments?

**Relation To Broader Scientific Literature:**

This paper fits well into the literature on causal inference and design uncertainty:
- Extends Aronow et al. (2014), Imbens & Menzel (2021) on causal bootstrap
- Goes beyond analytical copula assumptions by solving for optimal coupling numerically
- Relates to optimal transport and Frechet bounds literature
- Empirically complements recent work on balancing designs (Harshaw et al., 2024)

Positioning is clear and the novelty of integer-program-based causal bootstrap is well-motivated.

**Theoretical Claims:**

The paper contains multiple rigorous theoretical results:
- Lemma 2.1 guarantees feasibility of the IP under relaxed marginal constraints
- Theorem 4.1–4.3 prove asymptotic validity under unconfounded, conditionally unconfounded, and bounded confoundedness regimes
- Proofs rely on marginal balancing, inverse-probability weighting, and probabilistic bounds (Hoeffding-type inequalities)

The clarity of the mapping from copula constraints to linear conditions on binary indicator variables is a notable strength.

---

> ### Author Rebuttal · Authors · 2025-04-01
>
> We thank the reviewer for their thoughtful engagement with our work.
>
> ## Responses to Questions
>
> 1. Thank you for this question; please see the discussion of scalability in our response to Reviewer bwtT.
>
> 2. We currently discretize the continuous outcomes using the grid of observed outcomes, so that the discretization is lossless. A bigger issue related to discretization is that we require the marginal distributions of the imputed joint distribution to match the observed marginals. While the observed marginal will converge to its expectation asymptotically, in finite samples there can be a significant gap between the empirical sample and the marginal distribution (characterized by the DKW inequality). This finite-sample gap can be mitigated through the epsilon slack parameter; see Section A.5 for results.
>
> 3. This is a nice suggestion to improve efficiency of the IP. We have not considered it, but it would be worthwhile to investigate in future work.
>
> 4. If SUTVA holds, our approach covers clustered designs, since they are usually implemented to be probabilistic and unconfounded, and the estimators they are usually paired with are linear- or quadratic-in-treatment. It is not immediately clear whether our approach yields good coverage when spillover effects are present. We agree with the reviewer that proposing a bootstrap procedure in this setting is an interesting area for future work.
>
> 5. This is a great question, and we would be happy to provide more practical guidance. In the regime of feasibility, the smaller epsilon is, the tighter the upper-bound for the variance, but the stronger the coverage guarantees as provided by Theorems 4.1 and 4.3 and Corollary 4.2. Some practitioners may choose to contrast the upper-bound provided by the Integer Program against these theoretical guarantees to select epsilon. In our simulations, we solved our IP with epsilon = 0 (feasible since N_1 = N_0, cf. Lemma 2.1) and found the coverage to be acceptable. The adaptive or selection of epsilon is an interesting area for future work.
>
> 6. We did not fully understand this question. For the matched-pairs design, the optimal copula is the one computed by our IP. If the reviewer was asking whether there are other (non-IP) ways to construct a copula for this design in the literature, then we are not aware of any.
>
> 7. The causal bootstrap proceeds by first imputing a single (deterministic) outcome to each unit for the treatment it did not receive, which is why we need binary assignment of outcome to units (hence an IP). A convex relaxation would overestimate the variance of the worst-case copula, which could negate any power gains from the causal bootstrap. The same applies to dual OT (which would have the same objective value as the convex relaxation by strong duality). In the case of complete randomization, we found that an LP reformulation of the imputation problem was possible, but this was not the case for general treatment covariance matrices.
>
> 8. Our method is also a copula-based bound; we are optimizing the least favorable copula numerically instead of deriving it analytically. Our method differs from previous work because we allow a broader class of treatment assignment mechanisms. For some concrete comparisons: (1) Aronow et al (2014) derive the assortative copula as variance-maximizing for the difference-in-means estimator under complete randomization. Our technique recovers their copula in this specific setting. (2) Fan and Park (2010) estimate quantiles of the treatment effect distribution under effect heterogeneity, assuming iid treatment assignments. They show that the assortative copula is not sharp for measuring quantiles of the distribution of the treatment effects. By contrast, our method estimates the ATE and allows assignment mechanisms other than iid.
>
> 9. This is an interesting idea. The choice of estimands and estimators grows with the number of treatment options, but a priori, such an extension should be possible in most cases, and would be worthwhile to explore in future work. Beyond the asymptotic validity results which would need careful consideration, one would need to make sure that the estimator variance and potential outcome constraints under multi-valued treatment remain optimizable.
>
> 10. We are deeply committed to open-sourcing valuable code and research, and hope to do so here. The feedback from practitioners has been positive: many practitioners find bootstrap methods intuitive, and shy away from complex variance formulas, which was the impetus for this work. Anecdotally, we find many users of A/B testing platforms shy away from implementing sophisticated randomized designs because they fail to see strong variance improvements when paired with the standard (but often incorrect) bootstrap confidence interval construction methods. We hope that making the construction of correct confidence intervals easier for practitioners to implement will encourage them to adopt more sophisticated designs.

---

### Official Review · Reviewer_bwtT · 2025-03-25

**Overall Recommendation:** 4

**Summary:**

The paper presents a method that employs integer programming to maximize the variance of proposed estimators in randomized experimental design, addressing the issue of design uncertainty. It extends linear-in-treatment and quadratic-in-treatment estimators and generalizes assignment mechanisms using integer programming. The approach is built on a strong theoretical foundation.

**Claims And Evidence:**

Yes, the paper provides theoretical guarantees and empirical evidence to support the claims.

**Essential References Not Discussed:**

No

**Experimental Designs Or Analyses:**

Yes

**Methods And Evaluation Criteria:**

Yes, the authors validated the proposed approach via simulations on real data and compared it with three baselines.

**Other Comments Or Suggestions:**

No

**Other Strengths And Weaknesses:**

**Strengths:**
1. The paper tackles the important challenge of design uncertainty in experimental design and extends previous work by incorporating linear-in-treatment and quadratic-in-treatment estimators, along with new assignment mechanisms using integer programming.
2. The proposed approach is supported by asymptotic guarantees under different assignment mechanisms.
**Weaknesses:**
1. Computational complexity and scalability of the integer programming formulation are not thoroughly discussed, which may pose challenges for large-scale experimental designs.

**Questions For Authors:**

1. The authors establish the asymptotic validity of their approach under unconfounded, conditionally unconfounded, and individualistic assignments with bounded confoundedness. Could the authors elaborate on the assumptions underlying each of these assignment mechanisms?
2. Solving Integer Programming problems is computationally expensive as the problem size increases. Could the authors provide a more detailed discussion on the computational complexity and scalability of their approach?

**Relation To Broader Scientific Literature:**

Building on previous work, this paper extends to linear-in-treatment and quadratic-in-treatment estimators and introduces new assignment mechanisms using integer programming.

**Theoretical Claims:**

Yes

---

> ### Author Rebuttal · Authors · 2025-04-01
>
> We thank the reviewer for their careful and positive review of our paper. Below, we address the two questions raised.
>
> Q1:  Unconfoundedness implies that the treatment assignment is independent of any potential outcome, usually achieved through randomization. We distinguish between conditional and unconditional unconfoundedness, i.e. whether or not this independence requires conditioning on an observed covariate, though the literature does not always make this distinction. Strictly speaking, a stratified assignment is not unconditionally unconfounded and is instead conditionally unconfounded if the stratifying variable is correlated with the potential outcomes. In observational studies, we usually cannot eliminate confounders, and it is more common then to assume conditional unconfoundedness. An excellent discussion on the relationship between randomization and unconfoundedness is [1].
>
> Bounded confoundedness is not a standard defined assumption like unconfoundedness is. It typically arises in the context of sensitivity analysis and assumes that the magnitude of any confoundedness is limited within some range, i.e. one where treatment probabilities do not change drastically with remaining confounders. An individualistic assignment is one where the treatment assignment of one unit is independent from the treatment assignment of other units. For example, a Bernoulli randomized design is individualistic, but a completely randomized design, strictly speaking design is not (it is unconfounded however, so it is covered by Theorem 4.1).
>
> We would be happy to expand on these assumptions and include them in the paper if the reviewer would find it helpful.
>
> [1] Sävje, Fredrik. "Randomization does not imply unconfoundedness." arXiv preprint arXiv:2107.14197 (2021).
>
>
> Q2: Regarding IP scalability, we would first like to re-emphasize that our method is primarily motivated for small-sample experiments where design uncertainty dominates. Also, we note that given the results of a particular experiment, the IP only needs to be solved once. (In our simulations, the IP had to be solved multiple times for coverage and power analysis.) Nonetheless, we agree that for a general-purpose tool, scalability is important.
>
> The scalability of solving the IP depends on the sparsity of the treatment covariance matrix. This is why we see sub-quadratic scaling for Matched-Pairs in Table 1 (Appendix A.8). For the case of complete randomization where the off-diagonal entries to the covariance matrix are all equal, an LP formulation is possible as mentioned in Section 5.
>
> To scale the IP beyond hundreds of units, we see potential to apply relaxation or approximation algorithms, as also suggested by Reviewer XXTL.
>
> For a relaxation, we could relax the IP to an LP. The objective would then provide an upper-bound (i.e., an overestimate) to the worst-case variance, which can be used for Neyman-style confidence intervals (e.g., 1.96 * sqrt(variance upper bound)). However, the LP solution could not be used for the causal bootstrap because it might not be feasible for the IP.
>
> For approximation, on the other hand, we could apply some technique like LP rounding to obtain a feasible solution, but this would now underestimate the variance. However, if we can guarantee that we have a c-approximation (so that the solution objective is 1/c of the optimal, for c >= 1), then one could scale imputed unit outcomes by sqrt(c) to ensure the variance of the causal bootstrap distribution upper bounds the true variance. Developing LP rounding techniques for our IP is still an open question.

---

### Decision · Program_Chairs · 2025-05-01

**Decision:**

Accept (poster)

**Comment:**

This paper introduces a new method for quantifying design uncertainty in causal inference, particularly useful for small sample sizes or non-standard assignment mechanisms. Traditional bootstrap methods primarily address sampling uncertainty, while existing causal bootstrap techniques are often limited to completely randomized designs and simple estimators, such as the difference in means. The authors expand on causal bootstrap by employing integer programming to manage a wider range of estimators and assignment scenarios, including varied levels of confounding.

They validate their approach asymptotically and demonstrate its effectiveness through improved variance estimates and tighter confidence intervals in simulated geographical experiments using IMF data. This method allows for exact variance maximization without relying on closed-form copulas.

Overall, the proposed method contributes significantly to the literature on causal inference and design uncertainty. Given the importance of evaluating causal effects from confounded data, this work could have a substantial impact across various disciplines. However, since integer programming is generally computationally demanding, there are challenges in scaling the proposed method to larger datasets. Exploring efficient approximations for integer programs could be a promising future direction.